# Parallel within-host evolution alters virulence factors in an opportunistic *Klebsiella pneumoniae* during a hospital outbreak

Greta Zaborskytė ®[1,4], Karin Hjort ®[1], Birgitta Lytsy ®[2,5] & Linus Sandegren ®[1,3] ✉

Bacterial pathogens adapt to host niches because of selective within-host pressures, an evolutionary process that offers invaluable insights into host-pathogen interactions. Here, we retrospectively track the evolution of a single multiresistant *Klebsiella pneumoniae* clone in 110 patients over a 5-year nosocomial outbreak. We combine comparative genomics with phenotypic characterization of mucoviscosity, serum survival, iron utilization, biofilm formation, and infection potential in *Galleria mellonella* for all isolates. Strong positive selection within patients targeted key virulence factors. Notably, convergent evolutionary trajectories were dominated by reduced acute virulence and recurrent changes in iron uptake regulation, capsule and lipopolysaccharide, and enhanced biofilm formation. These phenotypic changes likely represent clinical niche adaptations, with some resulting in trade-offs during gastrointestinal colonization. This study underscores the dynamic nature of within-host evolution and its role in shaping virulence in opportunistic pathogens, even on short time scales.

Multidrug-resistant (MDR) *Klebsiella pneumoniae* is recognized as a critical priority pathogen by the World Health Organization[1]. The burdensome MDR phenotype is typically associated with *K. pneumoniae* of the classical pathotype (cKp), which interacts with the host opportunistically[2,3]. cKp frequently colonizes the gastrointestinal tract (GI) or nasal mucosa without causing disease[4,5] but can invade various tissue compartments due to underlying patient conditions such as immune suppression, indwelling medical devices, and antibiotic treatment[3,6]. Therefore, cKp infections are predominantly nosocomial, presenting mainly as urinary tract infections (UTI), pneumonia, or wound and soft tissue infections[3,7]. However, more pathogenic hypervirulent *K. pneumoniae* strains (hvKp) have evolved that cause invasive community-acquired infections even in individuals without any predisposing factors[8]. Capsule polysaccharides,

lipopolysaccharide (LPS), and siderophore-mediated iron uptake are the main virulence factors in *K. pneumoniae* in general[6]. Increased mucoviscosity, usually linked to overproduction of capsule, and elevated siderophore production were initially associated with hvKp[9–11], however, it has become evident that distinguishing hvKp from cKp is not straightforward and requires a combined evaluation of genotypes, phenotypes, and epidemiological metadata[8]. Currently, the presence of five genes (*iucA*, *iroB*, *peg-344*, *rmpA*, *rmpA2*) typically on pLVPK-like virulence plasmids is the most reliable, although not perfect, predictor for hvKp[10,12].

Bacterial interaction with the human host can lead to within-host selection of mutants better suited to specific host niches and/or with higher transmissibility[13–19]. The long-term fate of such mutants depends on their survivability in the particular niche and potential

---

[1]Department of Medical Biochemistry and Microbiology, Uppsala University, Uppsala, Sweden. [2]Department of Medical Sciences, Uppsala University, Uppsala, Sweden. [3]Uppsala Antibiotic Center, Uppsala University, Uppsala, Sweden. [4]Present address: Ineos Oxford Institute for Antimicrobial Research, Department of Biology, University of Oxford, Oxford, UK. [5]Present address: Department of Laboratory Medicine, Division of Clinical Microbiology, Karolinska Institute, Solna, Sweden. ✉e-mail: linus.sandegren@imbim.uu.se

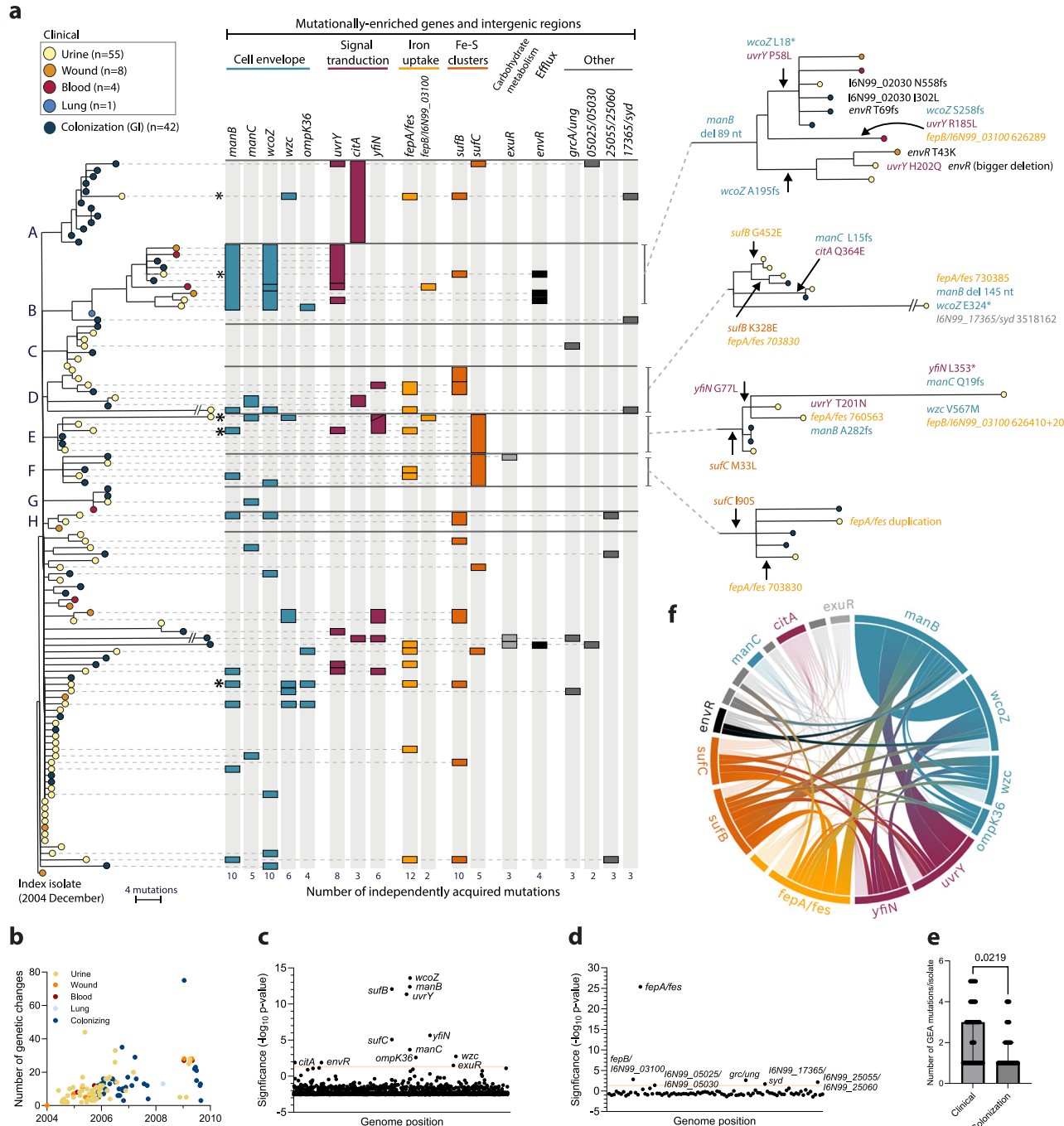

**Fig. 1 | Genomic evidence for adaptive evolution during the outbreak.**
**a** Maximum likelihood phylogeny of the 110 outbreak isolates rooted with the index isolate. Isolates are colour coded according to location of isolation as indicated by the key in the figure. Clusters of >2 isolates are assigned with letters A-H. The columns and boxes next to the tree show the presence of mutations in significantly enriched genes and intergenic regions. Colour coding indicates different functional groups as underlined at the top. Separate boxes indicate independent mutations. Zoomed-in clusters on the right illustrate the acquisition of enriched mutations in four of the clusters. These data are available for interactive viewing on Microreact (https://microreact.org/project/6BG86F2TGdDpZ4Q5xAvMGh). **b** Number of chromosomal genetic changes per isolate relative to isolation date. Colour coding of isolates according to isolation location as in 1a. Manhattan plots illustrating gene (**c**) and intergenic region (**d**) mutational enrichment analysis. Genes and intergenic regions that exceed a Bonferroni-corrected significance threshold of α = 0.05 (orange line) are named. **e** Number of GEA mutations per isolate with GEA mutations in clinical (*n* = 40) vs colonizing (*n* = 30) isolates. Bars show medians with 95% CI. The two groups were compared using Mann-Whitney test; two-tailed *p*-value is shown. **f** Chord graph illustrating the number of isolates carrying combinations of mutations between any of two enriched genes/intergenic regions. The combinations that were found in at least two isolates are shown with increased opacity. The same colour coding of functional groups as in (**b**). The graph was created using the Flourish Studio visualization tool. Source data are provided as a Source Data file.

trade-offs to their normal colonization and transmission routes, for *K. pneumoniae*, mainly the faecal-oral route. As named by Culyba and Van Tyne[20],"adapt-and-live" mutations combine the new trait with maintained pathogen transmission and are associated with globally disseminated clones. In contrast, "adapt-and-die" mutations help pathogens survive in specific host niches short term but limit their transmission to new hosts due to trade-offs in colonization potential or because the niche itself limits spread, thereby reducing their long-term

evolutionary potential[20,21]. The balance between colonization, infection, and transmission will, therefore, shape the evolutionary trajectories and what genetic changes prevail in the short and long term.

Evolutionary studies on *K. pneumoniae* virulence have mainly focused on the emergence of high-risk clones through genomic diversification by horizontal gene transfer and recombination[2]. In contrast, data on mutation-driven adaptations selected at the individual host level are scarce. Even though they are harder to identify, such adaptive changes are essential in shaping the interaction with the host and can negatively impact the infection outcome in individual patients[17–19]. Adaptive changes can also promote a more benign interaction with the host, and these will go undetected when focusing on high-risk clones.

The high genetic diversity of cKp[22] can obscure evolutionary trajectories in genome-wide association studies of unrelated bacterial isolates. Clonal hospital outbreaks allow for a more precise analysis of within-host evolution due to the common genetic origin of bacterial isolates, high sampling density, and a defined evolutionary time frame[23,24]. If the outbreak is large enough, signals of adaptation can be detected and evolutionary trajectories followed by analyzing repeated independent genetic changes, combinations of genetic changes, and convergent evolution of the same phenotypic traits.

Between 2005 and 2010, a clonal MDR *K. pneumoniae* outbreak occurred at Uppsala University Hospital (Sweden), affecting at least 320 patients through colonization or infection[25–27]. The outbreak clone belonged to sequence type (ST) 16, which has been linked to nosocomial outbreaks globally[28–30]. The first patient, identified in December 2004, had recently been treated in Tehran (Iran) before being admitted to the oncology ward at Uppsala University Hospital[25]. Epidemiologic investigations revealed that the spread at the hospital occurred via direct and indirect patient-to-patient transmission, rather than from a common environmental source[26]. The infection profile was typical for cKp, primarily UTIs among elderly patients with underlying conditions and risk factors[31]. The clone carried a 5 kbp cryptic plasmid (pUUH239.1) and a 220 kbp multidrug resistance plasmid (pUUH239.2) encoding resistance to β-lactams, aminoglycosides, tetracyclines, trimethoprim, sulphonamides, quaternary ammonium compounds, macrolides, and heavy metal ions (silver, copper, and arsenic)[27], but lacked any hypervirulence-associated genes.

In this study, we conducted a large-scale genomic and phenotypic analysis to investigate the within-host evolutionary trajectories of the *K. pneumoniae* clone in 110 patients during asymptomatic GI colonization and opportunistic infection. We identify strong positive selection that repeatedly acted on key virulence factors and demonstrate how these genetic changes translate into diverse adaptive phenotypes connected to survival in the host.

## Results

### Phylogenetic analysis of outbreak isolates

To explore genetic and phenotypic changes within the clone during the outbreak, we analyzed 110 clinical and colonizing isolates distributed throughout the five years (Supplementary Fig. 1). These isolates were obtained from individual patients at the first occasion of isolation. Clinical isolates ($n = 68$: urine $n = 55$, wound $n = 8$, blood $n = 4$, lung $n = 1$) denotes those where the patient presented symptoms of infection, while colonizing isolates ($n = 42$) were from rectal screening cultures starting from September 2006 as a measure to curb the outbreak[26].

We determined complete genome sequences for all isolates. Core genome comparison of globally sampled *K. pneumoniae* ST16 showed the Uppsala outbreak clone clustering with other European isolates (Norway, Germany, Italy, Hungary) with later isolation dates (see Swedish isolates in ref. 29). A maximum likelihood phylogeny of the 110 Uppsala isolates, rooted with the index isolate, showed a topology with a stepwise accumulation of individual mutations and eight

clusters (A-H) (Fig. 1a). The isolates accumulated 1 to 75 mutations (median 10) compared to the index isolate (Fig. 1b), totalling 653 unique genetic changes, including SNPs, insertions, and deletions, in 407 chromosomal genes and 90 intergenic regions (Supplementary Data 1 and 2). Genetic changes in the accessory genome mainly involved IS-element insertions and structural variations in the MDR plasmid pUUH239.2, driven by homologous recombination between IS26-elements as described previously[27]. Six isolates contained additional sequences matching different plasmids, and all isolates in cluster B had a 70 kbp integrative conjugative element inserted in a copy of tRNA-Phe and had lost the small plasmid pUUH239.1. No isolates contained additional resistance or virulence-related genes.

### Adaptive within-patient evolution repeatedly targeted key virulence factors

We analysed genetic changes for signs of selection during the outbreak. A nonsynonymous vs. synonymous substitution ratio (dN/dS) higher than one in a protein-coding gene is generally considered a sign of positive selection[32]. The overall dN/dS of all 407 mutated genes was 2.4, and genes with three or more independent mutations ($n = 20$) exhibited a strong positive selection signal with a dN/dS of 49.7 (Supplementary Data 3).

Gene enrichment analysis (GEA)[33] identified twelve genes and six intergenic regions with significantly increased accumulation of independent nonsynonymous mutations (Fig. 1c, d). These mutations were widely spread in the phylogeny, clearly showing multiple independent acquisitions during the outbreak (Fig. 1a). The mutationally overrepresented targets were associated with the main *K. pneumoniae* virulence factors – capsule, LPS, and iron utilization, including genes coupled to the KL51 capsule (*wcoZ* acetyltransferase, *wzc* tyrosine kinase), O3b-type O-antigen (*manB* phosphomannomutase, *manC* mannose-1-phosphate guanylyltransferase), two-component system BarA-UvrY (*uvrY* response regulator), the iron-sulfur (Fe-S) cluster synthesis SufBCD complex (*sufB*, *sufC*), and the *fepA/fes* intergenic region, encoding regulatory elements for the catecholate-siderophore uptake receptor FepA and the enterobactin esterase Fes. Other mutationally enriched genes involved the outer membrane porin *ompK36*, regulation of c-di-GMP turnover (*yfiN* diguanylate cyclase), citrate metabolism (*citA* sensory histidine kinase), hexuronate utilization (*exuR* transcriptional repressor), and efflux (*envR* transcriptional repressor).

In addition to the overrepresentation of individual mutations in these targets, 36 isolates had mutations in more than one enriched target co-occurring in the same isolate (Fig. 1a). Five isolates had independent combinations of mutations in five enriched targets, and all these isolates were from UTIs (indicated with stars in Fig. 1a). Overall, combinations of mutations in enriched targets were more common in clinical than colonizing isolates (Fig. 1e), suggesting possible niche adaptations to growth outside GI that require a combination of altered traits[34]. Certain combinations were repeatedly acquired in independent isolates, such as *manB-wcoZ-uvrY*, *manB-wzc-ompK36* or *fepA/fes-sufB*, *fepA/fes-sufC* as well as *manB-fepA/fes*, *manB-sufB* and *wzc-sufB* (Fig. 1f).

The temporal order in which the mutations occurred can indicate the evolutionary trajectories. Cluster B is a notable example where an 89 nt deletion in *manB* was followed by three independent mutations each in *wcoZ*, *uvrY*, and *envR*, along different branches of the cluster (Fig. 1a, highlighted clusters to the right). Inactivating mutations in *manB* and *wcoZ* and nonsynonymous changes in *uvrY* were also frequent as individual mutations in isolates throughout the phylogeny. Clusters D, E, and F also accumulated repeated mutations in GEA-enriched targets, especially *sufB/C* and the intergenic *fepA/fes* region. This convergent accumulation of genetic changes suggests an interdependence between some mutational targets in creating the phenotype under selection or accumulation of compensatory mutations. In

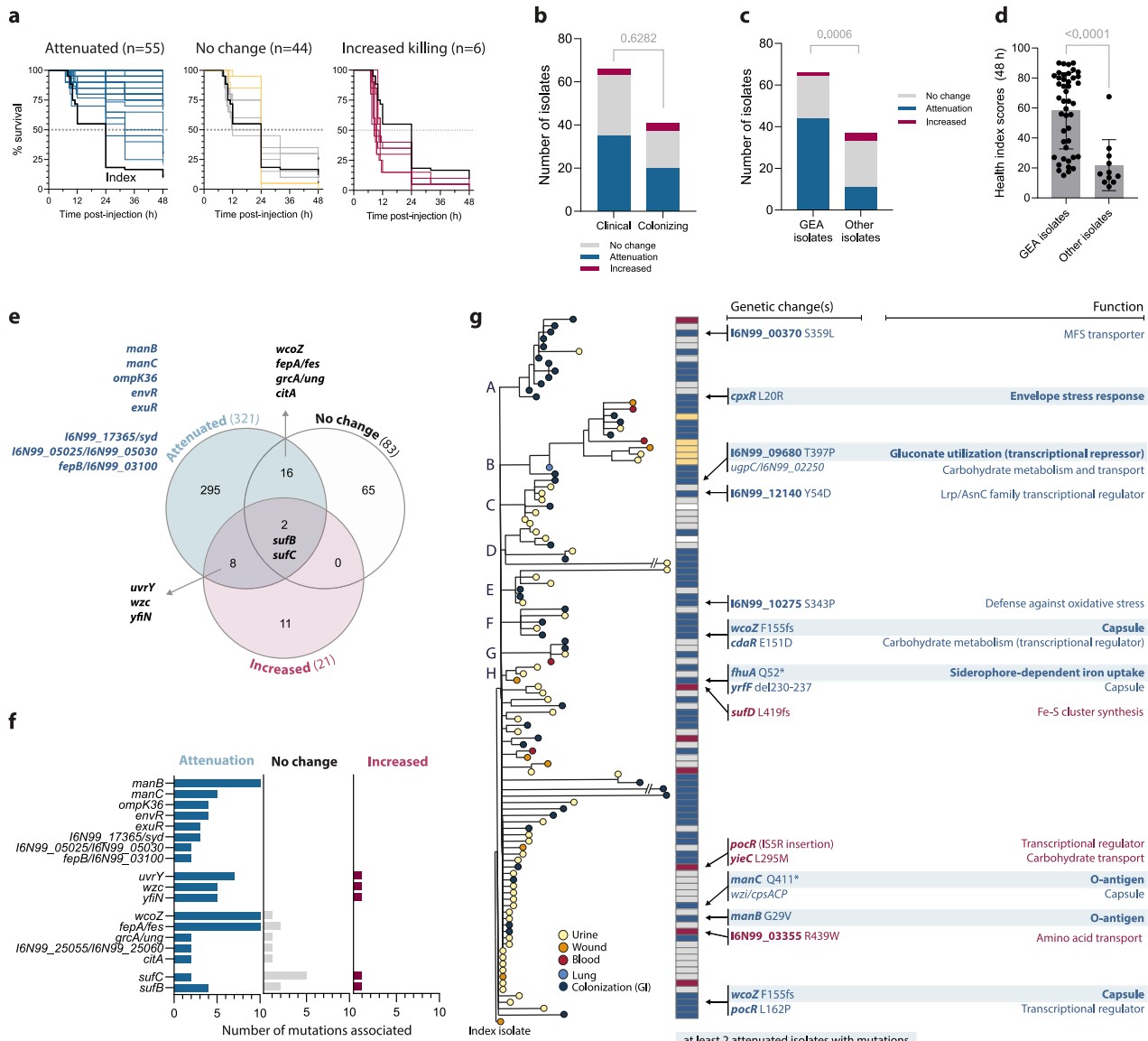

**Fig. 2 | Frequent attenuation of acute virulence is connected to mutationally enriched genes. a** Kaplan-Meier survival curves of *G. mellonella* larvae infected with bacterial isolates (*n* = 20 larvae for each isolate, 10 larvae per biological replicate of each isolate). Survival was assessed hourly 7–12 h post injection, then after 24, 32 and 48 h. Assignment of virulence groups was based on comparison to the index isolate (log-rank Mantel-Cox test). Two isolates (UTI and colonizing) with high variation after repeated testing were excluded from this grouping. Blue curves represent isolates with slower killing, grey isolates with unchanged killing, and red isolates with increased killing. Yellow curves indicate isolates with a different disease progression. **b** Distribution of virulence groups among clinical vs colonizing isolates. The comparison was done by Fisher's exact test; the two-tailed *p* value is shown. **c** Distribution of virulence groups among isolates that have GEA mutations (*n* = 66) vs isolates without GEA mutations (*n* = 41). The comparison was done by Fisher's exact test; the two-tailed *p*-value is shown. **d** Cumulative *G. mellonella* health index scores 48 h after injection for attenuated isolates with (*n* = 44) or without (*n* = 11) GEA mutations. Bars show medians with 95% CI. The higher health index score indicates decreased virulence, while 0 equals the death of all larvae. Comparison between groups was done by the Mann-Whitney test; the two-tailed *p*-value is shown. **e** Venn diagram of the mutated genetic targets (genes and intergenic regions) associated with different virulence profiles. Mutationally enriched genes and intergenic regions are listed. The full list of genes is available in Supplementary Data 4. **f** Prevalence of the respective GEA mutations among the virulence phenotypes. **g** Phylogenetic distribution of virulence phenotypes among the outbreak isolates and associated genetic changes. The indicated mutations were identified as the only differences from neighbouring isolates without any virulence change. Genes with at least 2 independent mutations associated with attenuation are highlighted. Source data are provided as a Source Data file.

agreement with this, functionally related mutations in non-enriched genes co-occurred together with the enriched ones: *citC* mutations together with changes in *citA*, *envR* mutations with diverse efflux-related mutations in isolates of cluster B (e.g., *oqxA*, I6N99_16545, *ramA*, *robA_TF*), *exuT* mutation with *exuR*, and *fepA/fes* mutants with other iron-related mutations (e.g., *fhuA*, *fepA*) (Supplementary Data 1).

Overall, the comparative genomic analyses indicate that genetic changes in several key virulence-associated processes underwent parallel selection during the outbreak and that

convergent evolutionary trajectories seem to involve a combination of altered traits.

## Acute virulence was repeatedly attenuated among isolates with enriched mutations

To assess how the identified genomic adaptations in virulence factors translate into phenotypic changes, we evaluated the ability of all 110 isolates to cause disease in the *Galleria mellonella* wax moth larvae infection model. *G. mellonella* has an innate immune system with both

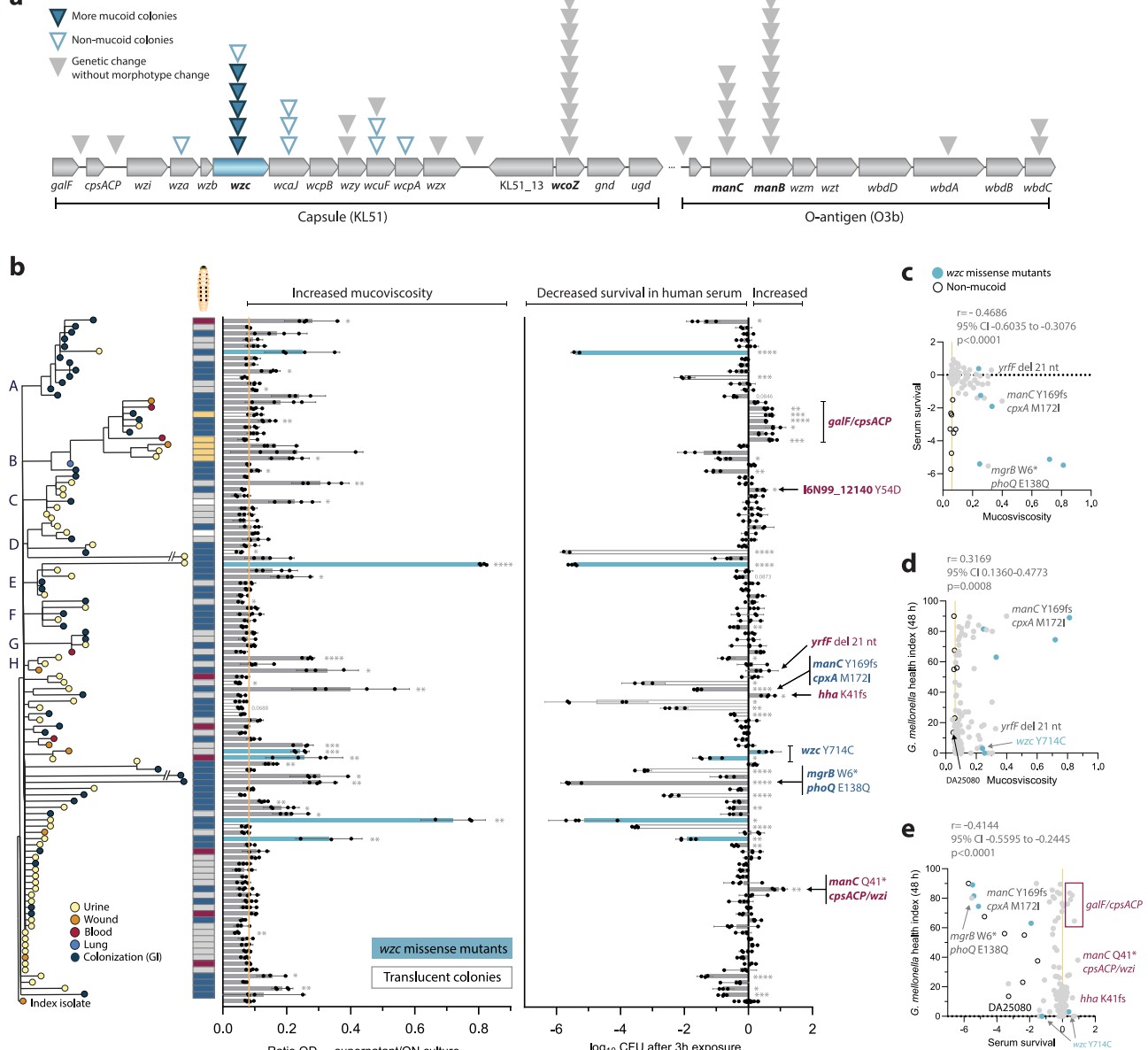

**Fig. 3 | Interconnection between mucoviscosity, survival in human serum and killing of *G. mellonella*. a** Mutations in *cps* (KL51) and O-antigen (O3b) loci and associated colony morphologies (morphotypes). Each triangle represents an independent mutation. KL and O-antigen type were determined using the Kaptive database. GEA genes are indicated in bold. **b** Sedimentation ratios (left) after low-speed centrifugation as a measure of mucoviscosity and bacterial survival in human serum (right), distributed according to phylogeny among isolates. Blue bars denote isolates with *wzc* missense mutations and white bars denote isolates with translucent colonies. Bars represent means of four biological replicates with 95% CI. Comparison of each isolate to the index was done by one-way ANOVA with Dunnett's T3 multiple comparison. Isolates where two-tailed $p < 0.05$ are marked with asterisks: *$p < 0.05$, **$p < 0.01$, ***$p < 0.001$, ****$p < 0.0001$. Boxes next to the phylogeny indicate the virulence group of each isolate, as in Fig. 2b, blue = attenuated, grey = unchanged, red = increased, yellow = different disease progression. Interconnections between **c** mucoviscosity and survival in human serum, **d** mucoviscosity and *G. mellonella* health scores at 48 h post injection, and **e** survival in human serum and *G. mellonella* health scores at 48 h post injection. Each dot represents mean values of the isolates based on data in (**b**). Blue dots indicate isolates with *wzc* missense mutations and clear dots - non-mucoid isolates. Pearson correlation coefficients (r) with 95% CI and two-tailed p-values are shown. Source data are provided as a Source Data file.

cellular and humoral responses similar to those of vertebrates[35,36] and is the preferred infection model for cKp, which do not perform well in mice models in contrast to hvKp[37–39]. We assessed larval survival and health status of the surviving larvae up to 48 h post-injection. Half of the isolates ($n = 55$) showed loss of acute virulence (attenuation), and six isolates exhibited increased virulence compared to the index isolate (Fig. 2a). Five isolates in cluster B displayed a unique infection pattern (although not identified as significantly different from the index in mortality rates), with larvae showing no signs of sickness up to 12 hours post-injection.

12 hours post-injection but dying rapidly afterwards (yellow curves in Fig. 2a). There was no difference in the distribution of virulence phenotypes between clinical and colonizing isolates (Fig. 2b), suggesting that loss in acute virulence did not eliminate bacteria from infection sites.

Isolates with GEA-enriched mutations were significantly overrepresented in the attenuation group (Fig. 2c). The virulence loss was also more severe among these isolates, as infected larvae had significantly higher cumulative health scores (Fig. 2d). Because of the

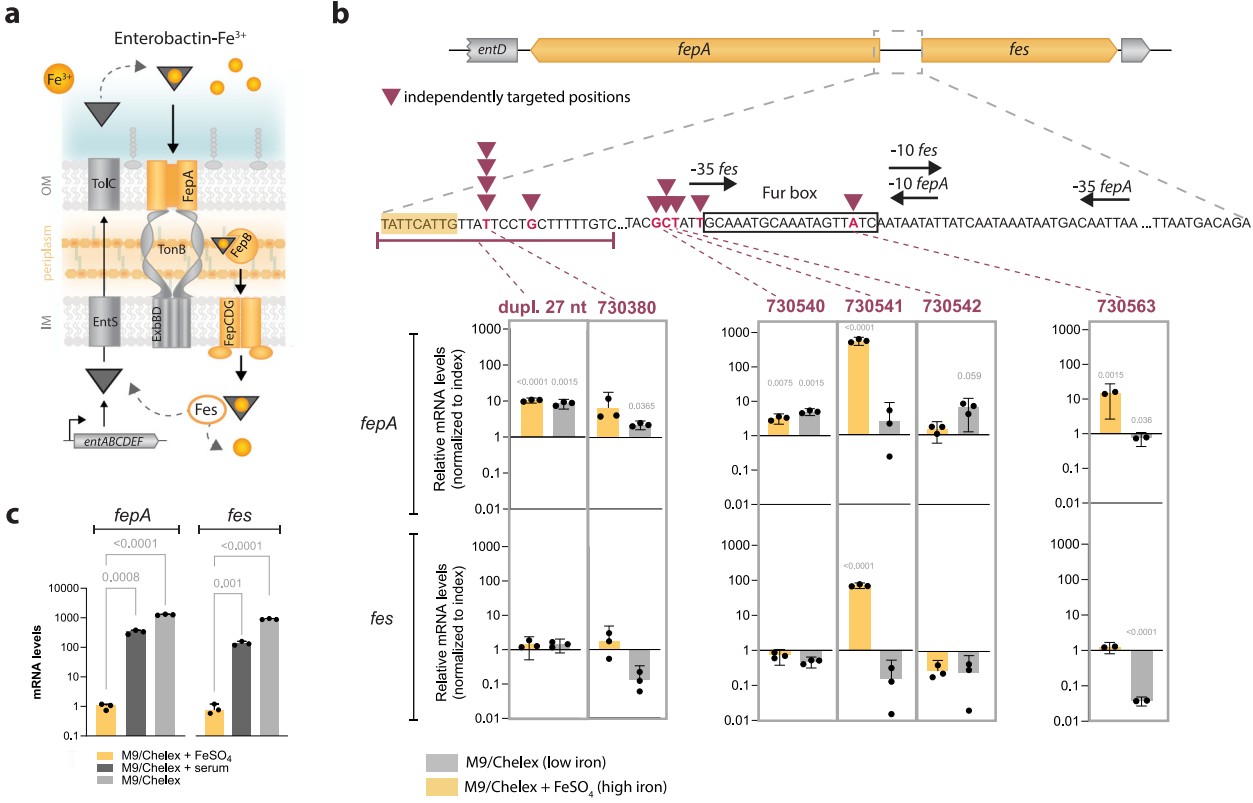

**Fig. 4 | Transcriptional changes in iron-dependent regulation of enterobactin uptake genes. a** Schematic overview of ferrienterobactin uptake by TonB-mediated active transport via FepA in *K. pneumoniae*. **b** Schematic representation of regulatory elements in the *fepA/fes* region and positions of mutations (noted in dark red) relative to the index isolate. mRNA levels for mutants (mean of *n* = 3 biological replicates with 95% CI) are relative to the index isolate for the respective conditions. Comparison to the index isolate was done by unpaired t-test with Bonferroni corrections for five comparisons; two-tailed *p*-values exceeding

significance (<0.05) are depicted above the columns. **c** Changes in *fepA* and *fes* mRNA levels in the index isolate under low and high iron conditions. mRNA levels (mean of *n* = 3 biological replicates with 95% CI) are relative to the housekeeping gene *glnA* and further normalized to M9/Chelex + FeSO₄. Comparison between different conditions for each gene was done by unpaired t-test with Bonferroni corrections for two comparisons; two-tailed *p*-values are depicted above the columns. Source data are provided as a Source Data file.

clonal origin of the outbreak, we could classify mutations likely associated with phenotypic changes based on genetic differences from the index isolate (in non-clustering isolates) or between neighbouring isolates in phylogenetic clusters (see Materials and Methods). In this manner, each nonsynonymous or intergenic tip and cluster mutation was assigned as likely to affect virulence or not. There was minimal overlap between mutated targets in the different virulence groups, with most unique hits among attenuated isolates. Eighteen genes and three intergenic regions were repeatedly mutated in at least two attenuated isolates, including the five GEA-enriched genes *manB*, *manC*, *envR*, *exuR*, and *ompK36* (Fig. 2e and Supplementary Data 4). Eleven genes, none of which were mutationally enriched, were uniquely associated with increased killing of *G. mellonella* but without overlap between isolates, reflecting diverging rather than converging genotypes associated with this gain of function, although the small number of isolates complicates such analysis. Several mutationally-enriched genes overlapped between different virulence groups (Fig. 2e), although mutations in *uvrY*, *wzc*, *yfiN*, *wcoZ*, and *fepA/fes* were strongly biased towards attenuation (Fig. 2f). This could suggest position-dependent mutational effects resulting in different phenotypes. For example, the only frameshift mutation in *uvrY* (N182fs) was linked to increased killing, while all seven missense mutations were linked to attenuation. Alternatively, the effects on virulence could be dependent on epistatic interactions with other mutations.

Mutations in genes identified as likely to influence virulence were often combined in the same isolate, complicating the distinction from

co-selection. However, 13 isolates had only one or two unique mutations linked to loss of virulence (*n* = 10) or to increased killing (*n* = 3), with six genes, including the enriched genes *manB*, *manC*, and *wcoZ*, mutated in at least two independent attenuated isolates (Fig. 2g). In addition to capsule and O-antigen, identified mutations were connected to cell envelope stress response, carbohydrate metabolism, MFS transporter, defence against oxidative stress, and an uncharacterized transcriptional regulator.

The genotypic and phenotypic data collectively show that loss of acute virulence, rather than increased virulence, was the adaptive trait in this clone during its spread in patients, and several of the mutationally enriched genes were associated with this.

## Diverging trajectories in mucoviscosity due to surface polysaccharide mutations explain decreased bacterial survival in human serum and virulence attenuation

Around half of the isolates (*n* = 53) had at least one and up to five nonsynonymous mutations in genes connected to surface polysaccharide synthesis, especially in the KL51 *cps* and O3b antigen loci that include the mutationally enriched *wzc*, *wcoZ*, *manB*, and *manC* genes (Fig. 3a, Supplementary Data 5). Some mutations resulted in colony morphology (morphotype) change, with isolates either forming translucent (non-mucoid) or more mucoid colonies despite the absence of a *rmpADC* locus in the outbreak clone. Increased mucoviscosity, which has been mostly linked to increased capsule abundance and/or altered polysaccharide chain length, corresponded with the

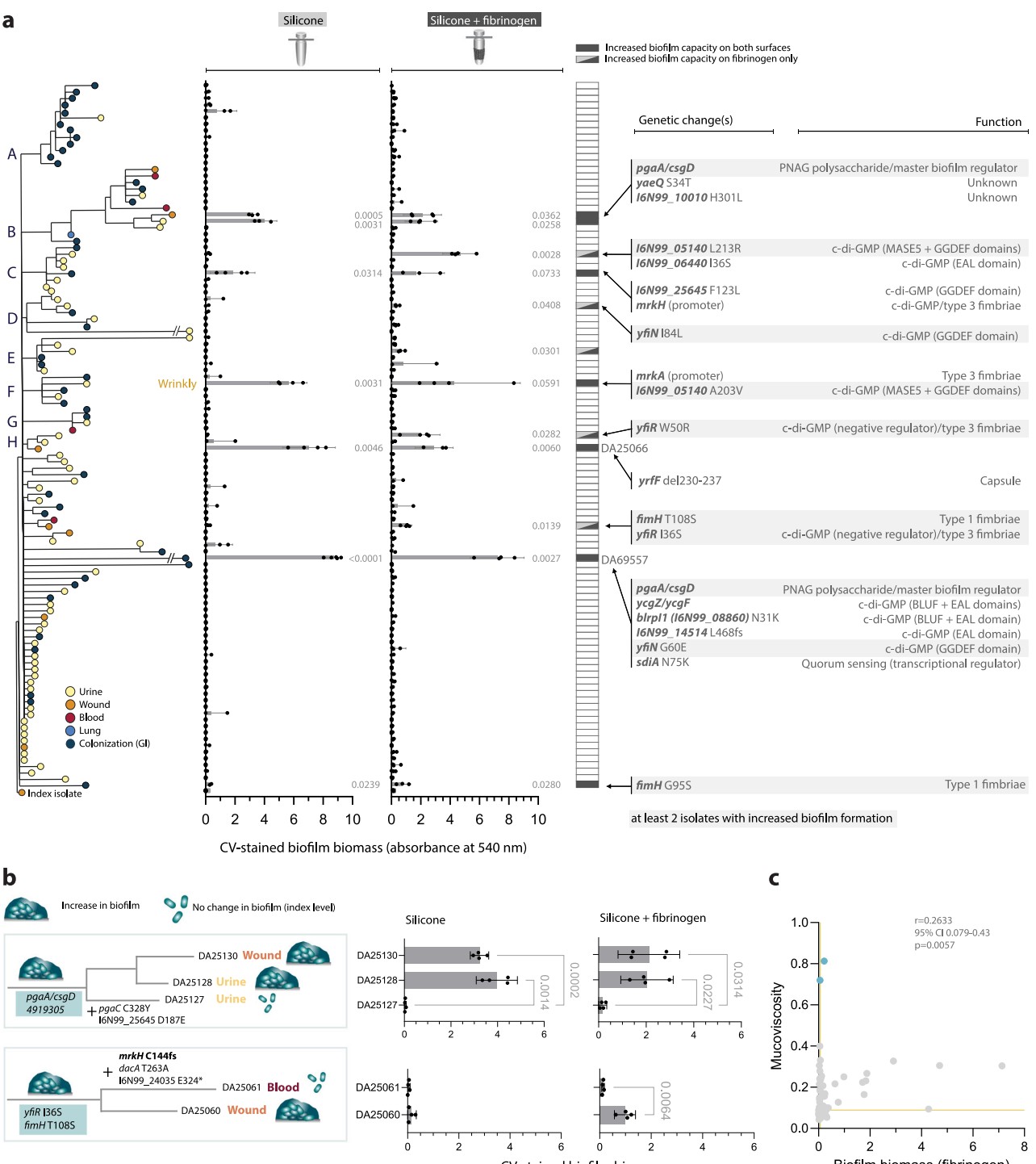

**Fig. 5 | Convergent evolution of biofilm formation capacity. a** Capacity to form surface-attached biofilms on silicone-coated pegs with or without fibrinogen coating, distributed according to phylogeny among isolates. Bars indicate means (*n* = 4 biological replicates with 95% CI) of total crystal violet (CV)-stained biomass of 48 h biofilms. Isolates where only one biological replicate showed an increase were not considered as these were likely spontaneous mutants that can be rapidly selected in this clone. Comparison to the index was done by one-way ANOVA followed by Dunnett's T3 multiple comparison; two-tailed *p*-values are shown. Highlighted genetic changes are most likely responsible for the increased biofilm phenotype based on nearest-neighbour comparisons. Genes with at least 2 independent mutations associated with biofilm increase are highlighted. **b** Biofilm evolution scenarios with consecutive genetic changes leading to increased or decreased biofilm formation. Means of four biological replicates with 95% CI are shown. Comparisons of DA25130 and DA25128 to DA25127 were done by one-way ANOVA with Dunnett's T3 post hoc, and DA25060 was compared to DA25061 by an unpaired t-test with Welch's correction; two-tailed *p*-values exceeding significance (<0.05) are shown. **c** Interconnection between acute virulence and survival in serum among isolates with increased biofilm formation capacity on silicone with fibrinogen. Blue dots indicate isolates with *wzc* missense mutations. Pearson correlation coefficient (r) with 95% CI and a two-tailed *p*-value is shown. Source data are provided as a Source Data file.

mucoid morphotype as quantified by sedimentation assay[40] (Fig. 3b). Seven isolates with increased mucoviscosity, all from urine or wound, had independent missense mutations in the GEA-enriched *wzc* gene, encoding a capsule co-polymerase and regulatory tyrosine kinase[41,42], or a 21-nt in-frame deletion in *yrfF*, encoding a repressor for the Rcs two-component system that induces capsule production[43] (Fig. 3b).

The non-mucoid morphotype, corresponding to capsule loss, was repeatedly acquired (*n* = 10) via IS-element insertions in *wzc* or *wza* (polysaccharide export protein), or frameshifts in the capsular glyco-syltransferases *wcaJ* or *wcuF* (Fig. 3a and b). Interestingly, many isolates maintained the mucoid index morphotype despite seemingly loss-of-function mutations in capsule genes such as *wcoZ*. In many *K. pneumoniae* strains, *manB/C* are located in the *cps* locus, and their inactivation abolishes capsule production, resulting in a non-mucoid morphotype[40]. However, in this clone, Kaptive[44] identified *manB/C* as part of the O-antigen locus, and since the KL51 capsule lacks mannose, the frequent mutations in *manB/C* are more likely to affect the synthesis of the O3b O-antigen that contains three mannose residues[45].

The capsule typically shields *K. pneumoniae* from innate immune attacks by the complement system[46]. A non-mucoid morphotype (capsule loss) is generally linked to impaired complement evasion, while increased mucoviscosity has been associated with complement resistance and more invasive infections[9]. We assessed survival in human serum for all 110 isolates and correlated this with mucoviscosity and genetic changes. Most isolates (*n* = 70) retained index-level survival, while 29 isolates became more sensitive to serum (Fig. 3b, right-hand graph). Surprisingly, more mucoid isolates with missense mutations in *wzc* (all from infection sites) showed the same increased sensitivity to serum as non-mucoid isolates, with up to a 5-log reduction in CFU/ml within three hours of exposure (Fig. 3b, c). This sensitivity was negated by the complement inhibitor poly-anetholesulfonic acid (Supplementary Fig. 2). In addition, decreased survival in serum despite increased sedimentation could be attributed to misregulation of the PhoPQ lipid A modification system (*mgrB* W6* and *phoQ* E138Q) and likely cell envelope perturbations (*manC* Y169fs and *cpxA* M172I). Improved survival in serum could be linked to a single intergenic *galF/cpsACP* mutation (cluster B), a combination of *manC* Q411* and intergenic *cpsACP/wzi*, a missense mutation in an uncharacterized transcriptional regulator I6N99_12140, or to an inactivating mutation in the hemolysin regulator *hha*.

We correlated mucoviscosity (Fig. 3d) and serum survival (Fig. 3e) with virulence in *G. mellonella*. Approximately half of the attenuated isolates showed decreased survival in serum (Fig. 3e). These included non-mucoid isolates and more mucoid *wzc* missense mutants, with a few exceptions: one UTI isolate (DA25080) without a capsule and two mucoid isolates with the *wzc* Y714C mutation retained virulence in larvae. Conversely, isolates with increased complement resistance via a combination of *manC* Q411* and intergenic *cpsACP/wzi* or *galF/cpsACP* mutations showed reduced killing of larvae. However, in three phylogenetic clusters, serum sensitivity via loss of capsule was acquired after virulence attenuation (Supplementary Fig. 3), and 24 attenuated isolates had unchanged serum survival. This suggests that even though changes in resistance against innate immune factors through altered capsule and LPS could frequently explain attenuation, other factors also contributed to virulence loss. There was a weak but significant negative correlation between *G. mellonella* killing and in vitro growth rates in rich medium or urine (Supplementary Fig. 4), indicating that in some isolates, general growth defects could be associated with reduced virulence, as shown before in *K. pneumoniae*[37].

## Selected intergenic *fepA/fes* mutations alter the iron-dependent regulation of enterobactin uptake genes

Iron sequestration is crucial for the growth and survival of *K. pneumoniae* within the host[47–49]. Multiple siderophore-dependent and independent systems were repeatedly mutated, including the *fepA/fes*

and *fepB*/I6N99_03100 intergenic regions, nonsynonymous mutations in *fepA*, inactivation of *fhuA* (hydroxamate-type siderophore receptor), and amplification of the ferrous iron uptake system *feoABC* operon (Supplementary Data 2). Moreover, iron acquisition-related mutations often co-occurred with mutations in *suf* genes, particularly *sufB* and *sufC*, which are involved in Fe-S cluster synthesis.

Enterobactin is the major catechol-type siderophore produced by *K. pneumoniae*, and ferrienterobactin is actively taken up via FepA (Fig. 4a). The *fepA/fes* intergenic region contains overlapping bidirectional promoters and a Fur (ferric uptake regulator) box that controls the expression of the FepA enterobactin transporter and the Fes enterobactin esterase in response to the extracellular iron levels (Fig. 4b). Since this intergenic region had accumulated 12 independent mutations among isolates, we assessed whether they led to changes in *fepA* and *fes* expression. The index isolate showed an intact regulation of expression of both genes depending on iron levels with high expression in chelated minimal M9 medium, low expression when supplementing with FeSO$_4$, and an in-between expression in serum-supplemented medium where most iron is bound to proteins (Fig. 4c). Mutations right upstream of *fepA* or within the Fur box either increased *fepA* expression (up to 10-fold) relative to index or disrupted the iron-responsive regulation, leading to constitutively elevated *fepA* and *fes* expression (up to 500-fold) even in high-iron conditions (Fig. 4b). These changes were unique to isolates with *fepA/fes* mutations (Supplementary Fig. 5).

We also measured how well all 110 isolates grew under iron-depleted and repleted conditions using FeSO$_4$ or inactivated human serum as iron sources. Although growth rates and yields decreased without iron, no clear connection was found between mutations and iron-dependent growth changes under these experimental conditions (Supplementary Data 6), possibly because partially redundant systems complement measurable deficits or because the genetic changes are involved in the scavenging of siderophores from other microbial species.

## Increased biofilm formation on catheter-like surfaces was selected via mutations affecting c-di-GMP turnover, fimbriae, and polysaccharide synthesis

Most clinical isolates were from UTIs, where urinary catheterization promotes biofilm growth of opportunistic uropathogens, including *K. pneumoniae*[50–52]. We quantified biofilm formation capacity for all isolates on silicone-coated surfaces with and without fibrinogen using the FlexiPeg device[53] to mimic catheter-associated biofilm growth. Fibrinogen, commonly deposited on urinary catheters during catheter-induced inflammation[54], is required for attachment by several uropathogens[55–57]. The index isolate was a poor biofilm former on these surfaces, although fibrinogen facilitated biofilm formation (Fig. 5a). Twelve isolates (nine from urine/wound) showed 100 to 3000-fold higher biofilm formation, with five isolates requiring fibrinogen coating to fully display the phenotype.

Isolates with increased biofilm formation had unique mutations in genes associated with the turnover of the intracellular signal molecule c-di-GMP and processes that are regulated by it, such as synthesis of type 3 fimbriae and poly-N-acetylglucosamine (PNAG), and these changes were often combined in the same isolate (Fig. 5a, right). Mutated c-di-GMP-associated genes included the mutationally enriched *yfiN* (diguanylate cyclase) and its negative regulator *yfiR*, but also uncharacterized EAL/GGDEF-domain genes (Fig. 5a, right). The isolate DA69557 had extremely high biofilm capacity[53], with mutations in at least four c-di-GMP-associated genes, including *yfiN* (Fig, 5a, right). Type 3 fimbriae were directly targeted via mutations in the promoters of c-di-GMP-dependent transcriptional regulator MrkH and the main subunit MrkA, and type 1 fimbriae via mutations in the tip adhesin FimH. An intergenic *pgaA/csgD* mutation (PNAG exporter/master biofilm regulator) was acquired independently in DA69557 and part of

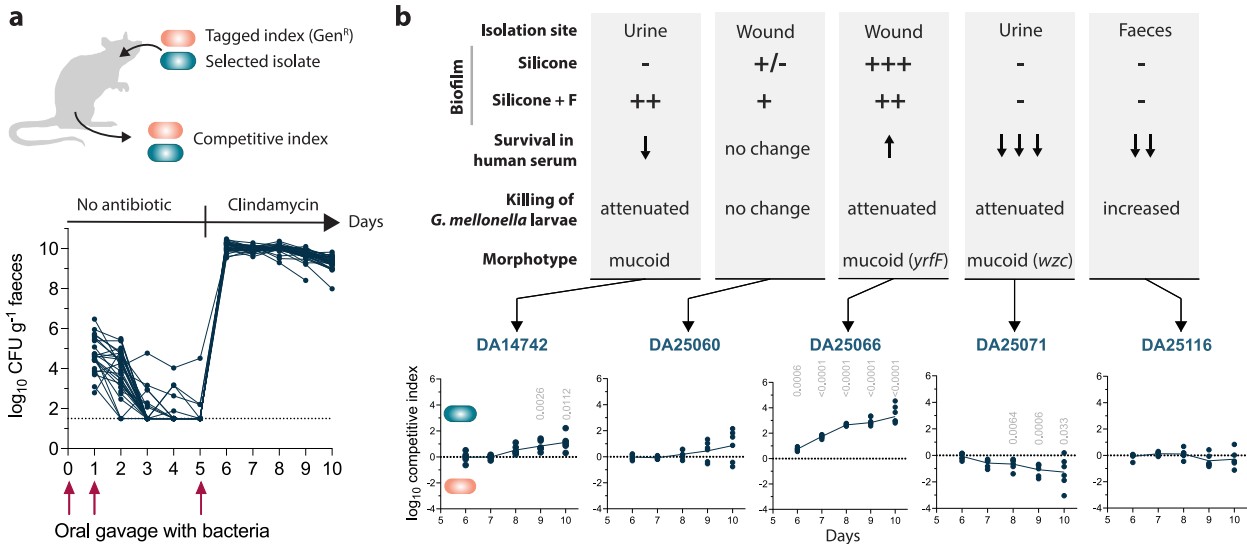

**Fig. 6 | Competitive potential for colonization of the gastrointestinal tract in mice. a** $3 \times 10^7$ CFU of 1:1 mixes of index and the respective test isolates were given by oral gavage at Day 0, 1, and 5. Overall CFU/g in faeces for each mouse ($n = 35$) is plotted during the 10-day experiment. On Day 5, clindamycin was added to the drinking water after a third oral gavage with bacteria. **b** Top panel illustrates phenotypes for each isolate. The lower panel displays competitive indices during days 6-10 of gastrointestinal colonization. Competitive index >0 means an advantage of the tested isolate versus the index isolate DA14734, <0 − disadvantage. Dots show the competitive index in individual mice ($n = 6$ per competition), and the trendline connects the means. Comparisons to control competition on each day were made by unpaired two-tailed t-test; Bonferroni corrected (5 comparisons) two-tailed $p$-values exceeding significance ($p < 0.05$) are shown above datapoints. Source data are provided as a Source Data file.

cluster B. The biofilm increase in wound isolate DA25066 was due to an already mentioned 21 nt deletion in *yrfF* increasing capsule production.

In most cases, increased biofilm capacity evolved as an end-point adaptation that was not transmitted further (Fig. 5a). However, the blood isolate DA25061 lost the increased biofilm phenotype on fibrinogen due to an inactivating mutation in *mrkH*, compared to the closely related wound isolate DA25060 (Fig. 5b). Similarly, a mutation in the essential PNAG synthesis protein PgaC reverted the biofilm phenotype to the index level on both surfaces in part of cluster B, which otherwise showed increased biofilm capacity due to the intergenic *pgaA/csgD* mutation.

One isolate with increased biofilm capacity showed a wrinkly/rugose colony morphology that is rare for *K. pneumoniae* but associated with enhanced biofilm in other bacteria via elevated c-di-GMP[58–61]. There was a tendency for isolates with higher biofilm capacity to have increased mucoviscosity relative to the index, and none of the non-mucoid isolates showed an increase in biofilm. However, most mucoid isolates, including those with *wzc* missense mutations, did not show increased biofilm formation on these surfaces (Fig. 5c). Production of other extracellular polysaccharides, such as PNAG or cellulose, might instead reduce sedimentation in isolates with increased biofilm formation. Although there was no correlation between biofilm capacity and serum sensitivity or virulence in *G. mellonella*, most biofilm formers had reduced acute virulence (Supplementary Fig. 6). Only the 21 nt deletion in *yrfF* could be attributed to mediate increased mucoviscosity, attenuation in *G. mellonella*, and greatly enhanced biofilm formation. In other cases, attenuation in larvae and an increase in biofilm formation were acquired by different mutations, illustrating separate evolutionary trajectories.

### Increased biofilm capacity but not *wzc*-mucoviscosity gives an advantage during GI colonization in mice

Since intestinal colonization is the primary reservoir for spreading to new hosts and precedes infection by cKp strains[5,62,63], we tested how a set of the isolates with distinct evolved phenotypes competed pairwise against the index isolate during colonization of the GI tract of mice. As for most cKp[64], colonization was not stable with the natural GI

microflora present for any of the competition pairs, indicating that the outbreak clone is not a super-colonizer under these conditions (Fig. 6a). Supplementation of the mice´s drinking water with clindamycin, which significantly disrupts the GI microbiota and enables long-term colonization with *K. pneumoniae*[64], led to an increase in bacterial load with above $10^9$ CFU/g faeces, sustained until the end of the experiment. This agrees with the previously observed transient super-shedder phenotype promoted by antibiotic treatments in mice and leading to enhanced transmission of *K. pneumoniae* from GI[65].

Three isolates had a selective advantage over the index isolate during GI colonization (Fig. 6b). They had different *G. mellonella* virulence and serum survival, but all had increased capacity to form biofilms. The wound isolate DA25066, which had the highest biofilm formation capacity and increased mucoviscosity due to a 21 nt deletion in *yrfF*, had the most substantial competitive advantage and rapidly outcompeted the index isolate. In contrast, the UTI isolate DA25071, where increased mucoviscosity was mediated by a missense mutation in *wzc*, was worse at colonizing GI than the index isolate. The colonizing isolate DA25116 remained neutral in competition. Collectively, these results illustrate that several of the within-host evolved phenotypes were combined with an increased colonization potential, while the mucoid *wzc* mutant isolated from a UTI represents a likely adaptation to the infection site with a trade-off in GI colonization ("adapt and die").

## Discussion

Understanding the evolutionary dynamics of bacterial populations within hosts is central to infection biology. This study provides insights into the within-host evolution of *K. pneumoniae* during a prolonged hospital outbreak. We observed significant genetic and phenotypic diversity among isolates, driven by mutations in key virulence factors and regulatory pathways. These findings highlight the adaptive potential of *K. pneumoniae* in response to selective pressures within the host environment.

As an opportunistic pathogen with distinct colonization and infection sites, *K. pneumoniae* represents an interesting case of adaptive evolution that can be explained using the "source-sink" model[21].

Stable maintenance and transmission occur mainly from GI-colonization (source) that precedes transient opportunistic colonization at infection sites (sinks). Mutants found at infection sites could also be pre-selected already during GI colonization if they have an advantage at both body sites. Converging adaptations may therefore result from selection pressures during GI colonization, infection, or both, most likely with stronger selection in the suboptimal sink locations[21]. While rich sugar-based carbon sources and competition with other microbiota prevail in the intestines, the urinary tract, from which most clinical isolates came, is typically nutrient-poor, dominated by amino acids, limited iron, and fierce host defences[66]. The latter fits well with the mutationally enriched genetic targets identified here and they were often combined in the same isolate, with a bias towards clinical isolates. Some combinations were functionally connected. For example, genetic changes in iron uptake often co-occurred with mutations in the Fe-S cluster assembly complex, induced under oxidative stress and iron starvation[67]. Mutations in *sufB* have been linked to increased iron binding[68], and variations in *suf* genes were found to be positively selected in the global *K. pneumoniae* outbreak lineages ST258[68] and ST147[69]. Similarly, combinations of mutations in surface polysaccharides and their regulation (e.g., *wcoZ*, *manB/C*, *wzc*) and the capsular response regulator UvrY[40] were frequent in this clone. Functionally distinct combinations (e.g., *manB* with *fepA/fes*) instead illustrate the diversity of selection pressures in specific host niches.

Virulence evolution in *K. pneumoniae* is often discussed in the context of increased acute virulence, such as hypervirulent isolates. However, the predominant evolutionary trajectory in this outbreak clone was reduced evasion of innate immunity and attenuation in acute virulence. Many attenuated isolates were from infection sites, suggesting that they still cause typical cKp nosocomial infections, and the convergent mutation patterns suggest that this phenotype is selected there. This may represent adaptation to a more chronic state, similar to other opportunistic pathogens[70], possibly through reduced immune recognition. Frequent inactivating O-antigen (*manB/manC*) mutations support this evolutionary trajectory, similar to adaptations seen in *Burkholderia* in CF lungs and during experimental biofilm evolution[13,60]. Loss of O-antigen due to its immunogenicity has been reported during late stages of chronic cystic fibrosis (CF) lung infections[71] and recently among carbapenem-resistant *K. pneumoniae* with KL47 and KL64 capsular types[72]. The largely uncharacterized capsule acetyltransferase gene *wcoZ*, which was independently mutated 10 times among isolates, seems to correspond to *orf13* in hypervirulent KL57 isolates, where acetylation was linked to increased immunogenicity without affecting capsule abundance[73]. Here, selection for decreased immune recognition could instead explain the exclusively inactivating mutations in *wcoZ*.

The capsule locus of *K. pneumoniae* is a known recombination hotspot[2] with over 130 different described capsule locus (KL) types[74]. Recent studies observed bimodal diversity in capsule phenotypes via inactivating mutations in initial glycosyltransferases (*wcaJ/wbaP*) and *wzc*-mediated increased mucoviscosity[72,75,76]. However, reported phenotypes vary, possibly due to strain background differences. For example, Cheng et al. linked capsule loss in ST258 to increased serum sensitivity and attenuation in a murine bloodstream infection model[76], while Hu et al. saw increased serum survival but increased susceptibility to phagocytosis due to increased expression of other surface polysaccharides, like O-antigen, in carbapenem-resistant ST11 isolates[72]. Ernst et al. associated *wzc*-mediated mucoviscosity with macrophage resistance and higher virulence in zebrafish larvae and murine UTIs for ST258[75]. Intriguingly, here, both more mucoid and capsule-loss isolates displayed severe serum sensitivity and were attenuated in *G. mellonella*, except for two isolates with a SNP in the C-terminal tail of the Wzc kinase. Both capsule loss and increased mucoidy were detected in infection sites across all these studies, but without a consensus on niche specificity. While Ernst et al. suggested *wzc* mutant to be associated with

bloodstream infections and capsule-loss with UTIs[75], Cheng et al. observed both types arising during bloodstream infections[76], and Hu et al., did not see any difference in capsule loss frequency across body sites[72]. We found capsule-loss in UTI and colonizing isolates, while *wzc*-mediated mucoid isolates were from UTIs and wounds. Supporting their connection to UTIs, urine was recently shown to modulate the capsule and maintain the mucoid phenotype of *K. pneumoniae wzc* mutants across various capsule types[77]. This phenotypic variation and possible linkage to host niche adaptations across different strains and capsule backgrounds merit further studies.

The most mutationally enriched intergenic region, *fepA/fes*, was associated with significant alterations in enterobactin uptake regulation and responsiveness to extracellular iron levels. In accordance with this, adaptive changes in siderophore uptake rather than synthesis seem to be critical during infection[78]. Intergenic regions related to iron acquisition were also under positive selection in carbapenem-resistant *E. coli* (ST410)[79]. Notably, mutations predicted to adversely affect iron acquisition did not reduce growth in low-iron conditions, suggesting a redundancy in how *K. pneumoniae* obtains and uses iron, as shown previously for ST258, where a mutation in the conserved enterobactin export protein led to higher expression of hemin transport[48,49]. These changes might be too subtle to be captured in growth assays or could require an environment better resembling the in vivo situation, e.g., with the presence of xenosiderophores that can be scavenged from other microorganisms. FhuA and FepA siderophore uptake proteins also serve as receptors for phages and bacteriocins[80,81]; thus, their loss could be selected as protection, separate from iron metabolism. In agreement with this hypothesis, one isolate with a *fhuA* mutation also had an inactivating mutation in *sbmA*, encoding the transporter for microcin-type bacteriocins.

Several isolates independently acquired enhanced biofilm-forming capacity on catheter-like surfaces, primarily via c-di-GMP-associated changes and regulatory or structural fimbrial alterations. Both type 1 and type 3 fimbriae are important for UTI pathogenesis, but type 3 fimbriae typically mediate attachment on abiotic surfaces like catheters[82,83]. Adaptive changes in the FimH tip adhesin have been reported both in *Escherichia coli* and *K. pneumoniae* from UTIs[84–87]. Increased biofilm formation is a long-term adaptation seen in various bacterial pathogens[60,88], including via mutations in the diguanylate cyclase YfiN and its negative regulator YfiR that were also associated with biofilm increase here. Unlike studies assessing biofilms on plastic microtiter plates[72,75], we did not observe a link between capsule loss and increased biofilm formation on silicone surfaces. Overall, it seems that biofilm evolution represents a separate trajectory from capsule and the associated virulence changes in this clone. The only mutation mediating multiple phenotypic changes, that is attenuation in *G. mellonella*, increased mucoviscosity, biofilm formation, and GI colonization, was the 21 nt in-frame deletion in *yrfF*. Previously, a 12 nt deletion in *yrfF* has been reported in *E. coli* isolated from keratitis[89], and an IS insertion in the *yrfF* promoter was selected during serial passaging in macrophages[90].

In this study, we performed genotype-phenotype associations by comparing isolates differing by single mutations or enriched mutational combinations. Due to the numerous individual mutations, combinations of mutations, and the difficulty in engineering clinical MDR isolates, detailed genetic reconstruction analyses will be conducted in follow-up studies to further understand interdependences and epistatic interactions between mutations and the genetic background. Additionally, while we have focused on some phenotypes important for survival in the host, this is only a subset of functional effects that could be connected to the genetic changes, and more functions could be covered in extended phenotype analyses.

In conclusion, our findings demonstrate the significant impact of integrating genomic analysis with phenotyping of clonal bacterial isolates in elucidating the evolution of a key nosocomial pathogen

during a large-scale hospital outbreak. We hope this dataset will be a valuable resource for understanding the within-host evolutionary patterns of opportunistic *K. pneumoniae*, the mechanisms underlying these adaptive changes, and their impact on pathogenesis.

# Method

## Bacterial sampling and growth

The isolates from the outbreak were collected from routine samples at Uppsala University Hospital between 2004 and 2010, each from a different patient, and only on the first occasion was it isolated, scraped directly from the isolation plate, and frozen at −80 °C. In total, 320 patients were identified with the clone by PFGE at the time of the outbreak, but unfortunately, no isolates except the ones gathered for this study were kept at the hospital, making it impossible to include additional isolates in the study in retrospect. Isolates were randomly included in this study to cover the entire outbreak duration and to represent the clinical and colonizing isolates equally. All isolates were included at times with low incidence, and a proportion of isolates were randomly chosen from time points with high incidence (Supplementary Fig. 1). For included isolates, 45% ($n = 50$) of the patients were male, and 65% ($n = 60$) were female. The median age was 80 years (range 23–100), and most patients had several co-morbidities and frequent visits to the hospital with treatment in multiple wards[26]. Isolates were propagated to a minimum to avoid selection in vitro. A summary of all isolates and associated genomic and experimental data is available as an interactive Microreact project (https://microreact.org/project/6BG86F2TGdDpZ4Q5xAvMGh).

## Whole-genome sequencing of outbreak isolates

The index isolate DA14734 was sequenced with long-read whole genome sequencing using the Pacific Biosciences platform (PacBio, Menlo Park, CA). DNA was prepared from overnight cultures using the Qiagen Genomic-Tip 100, according to the manufacturer's instructions. PacBio sequencing was performed at Uppsala Science for Life Laboratory on an RSII system with one SMRT cell per genome. In addition, the DA14734 genome was also sequenced with short-reads using MiSeq (Illumina, San Diego, CA) with two times 300-bp paired-end read lengths using the Nextera XT DNA library preparation kit (Illumina, San Diego, CA), according to the manufacturer's instructions. From the index isolate, a reference genome sequence was assembled with a combination of the PacBio reads and MiSeq reads. Assembly was performed with the CLC Genomics workbench v.20 (CLC Bio, Qiagen), including the Microbial Genomics Pro Suite module. The additional 109 isolates from the outbreak were also whole-genome sequenced with the Illumina technology, and the sequences were compared to the reference genome by reference assembly using the CLC Genomics Workbench. Variant detection was performed using CLC Genomics in-built detection modules for single-nucleotide polymorphisms (SNPs), insertions/deletions, and structural variations. All sequencing data have been deposited at NCBI under the bio project number PRJNA857654.

## Phylogenetic analysis

Variations in the chromosomal sequences were used to construct a Maximum likelihood phylogenetic tree using the CLC Microbial Genomics Module Epidemiology function. An initial Maximum Likelihood Phylogeny was created with a Neighbour-Joining General time reversible nucleotide substitution model and rooted on the index isolate. All identified genetic changes were then traced and mapped on the resulting tree, and branch lengths were adjusted to represent the integer genetic distances between nodes.

## Bioinformatic analysis of genetic changes

The dN/dS ratio was calculated for groups of mutated genes with the same number of independent mutations by the method of Nei and Gorobori[91] without correction for possible multiple changes in the same position due to the short time span of the outbreak. Identification of biological functions and grouping of genes was done by combining manual BLASTp[92], STRING[93], and EggNog 5.0[94] analysis and literature searches. Gene enrichment analysis was performed for both annotated open reading frames and intergenic regions using the approach and scripts presented by Elgrail et al.[33] with *p*-values adjusted for multiple comparisons using the Bonferroni correction. Capsule synthesis and O-antigen loci for the clone were determined by the Kaptive software[44].

## Genotype-phenotype connection analysis

Because of the clonal origin of isolates, the genetic basis for phenotypic changes was determined by first identifying genetic changes unique to each isolate as opposed to the genetic changes present in clusters (i.e., transmitted to multiple isolates). Within each cluster, isolates with phenotypic differences were compared to identify non-synonymous and intergenic genetic changes that are likely associated with the phenotype. In cases with more than one possible mutation, we checked whether (1) any gene was mutated more than once in isolates with the same phenotype, (2) whether mutations in the gene were present or absent in any other isolates displaying a different phenotype, (3) whether mutations were associated with any known virulence factors. If all isolates from a certain cluster displayed the same phenotype, both cluster and tip mutations were considered as likely causes of the phenotype.

## Exponential growth rate measurements in MHB II and urine

Overnight (O/N) cultures were diluted 1000-fold in cation-adjusted Mueller-Hinton Broth (MHB II, Sigma-Aldrich), and 300 µl were transferred to 100-well honeycomb plates. Five biological replicates were used for each isolate. The plates were incubated in a BioScreen C (Oy Growth Curves Ad Ltd) for 16 h at 37 °C with shaking, with $OD_{600}$ measurements taken every 4 min. For experiments with urine, morning urine was collected from a healthy female volunteer on six different days. The urine was centrifuged (3800 x *g*, 10 min), sterile filtered (0.2 µm pore size), and aliquots were frozen at −20 °C. For experiments, frozen aliquots from different days were thawed, centrifuged again (3800 x *g*, 10 min), and pooled together. Exponential growth rates were analyzed using the R script-based tool BAT 2.0[95].

## Biofilm formation

The capacity to form surface-attached biofilms was determined on the FlexiPeg biofilm device[53] with silicone-coated pegs. To further coat the pegs with fibrinogen, the autoclaved peg lid was incubated in a flat-bottom 96-well plate with 200 µl of 100 µg/ml fibrinogen from human plasma (product number F3879, Sigma-Aldrich, in 0.9% NaCl or PBS) at 4 °C overnight. Approximately $10^5$ CFU in a total volume of 150 µl were added to a 96-well plate, where the autoclaved peg lid was inserted, and biofilms were allowed to form statically in BHI for 48 h at 37 °C with a medium change after 24 h. To quantify the total biomass[53], the pegs with biofilms were washed three times with PBS, dried at 37 °C for 30 minutes, and then stained using 0.1% crystal violet (CV) (Sigma-Aldrich) at room temperature for 20 minutes. Pegs were then washed 3-4 times with PBS to remove unbound stain and dried for 20 minutes at room temperature. After dissolving the stain in 200 µl of 10% acetic acid, the absorbance (540 nm) was measured. All isolates were tested with at least 4 biological replicates (inoculating cultures started from independent colonies) with independent assays on silicone and silicone with fibrinogen.

## Bacterial survival in human serum

O/N cultures were grown in LB and diluted 100-fold in PBS. A final dilution of -$10^5$ CFU was mixed with human serum (Sigma-Aldrich, H6914 from male AB clotted whole blood, USA origin, sterile-filtered)

in a total volume of 200 μl in a 96-well plate. Due to the natural batch variation of human serum, serum was diluted up to 40% when needed to achieve a similar killing effect relative to the index and sensitive control isolates. For negative controls, serum was heat-inactivated for 30 min at 56 °C. To confirm that the killing effect was due to the complement system, a control with the complement inhibitor poly-anetholesulfonic acid (Sigma-Aldrich) was done at a final concentration of 100 μg/ml (Supplementary Fig. 2). CFU counts on LB agar (LA) were performed at timepoint 0 h and after 3 h static incubation at 37 °C.

### Virulence testing in *Galleria mellonella* larvae model

Research-grade wax moth *G. mellonella* larvae TruLarv™ were purchased from BioSystems Technology Ltd (UK). O/N cultures (BHI) were diluted in PBS, and 10 μl containing $10^4$ CFU were injected via the first right proleg using a 25 μl Hamilton 7000 syringe (Model 702 RN, 22S gauge needle). A dose of $10^4$ CFU was chosen because it led to the most consistent results between biological replicates and the best resolution between different virulence levels. For each strain, at least two biological replicates (cultures started from independent colonies) were injected into 10 larvae each. For each experiment, 10 larvae were injected with 10 μl sterile PBS to account for possible injection trauma, and 10 larvae were used as non-injected controls. Aliquots from dilutions were plated for CFU counts to confirm the correct infection load. Larvae were incubated at 37 °C for 48 h with an hourly assessment of death during 7–12 h post-injection, followed by monitoring after 24 h, 32 h, and 48 h. Larvae were considered dead when not moving after stimulation with a sterile pipette tip, which usually coincided with melanization. At the end of the experiment, live larvae were given health index scores[96], and a cumulative score (0-90) was determined by adding up scores of individual larvae for each replicate. Survival analysis was performed using the Kaplan-Meier method and analyzed by the log-rank (Mantel-Cox) test in GraphPad Prism version 9.2.0 for macOS (GraphPad Software, San Diego, CA, USA). All isolates were categorized relative to the index isolate DA14734. Two isolates, DA25084 and DA25063, that showed high variation after repeated experiments were not placed in any virulence category.

### Construction of a genetically marked index isolate for mice experiments

The index isolate DA14734 was transformed with the pSIM10-hygro and selected on LA with hygromycin (100 mg/L) at 30 °C. pSIM10-hygro carries the temperature-inducible λ Red recombineering system and is derived from the original pSIM9[97]. The aminoglycoside resistance gene *armA* was PCR amplified using Phusion High-Fidelity DNA Polymerase (Thermo Fisher Scientific Inc.) with primers introducing a 40 bp homology region directly outside *galK* for in-frame replacement (primers: 14734_armA_galK_F1 GCGCCGTCAGCGACGTCCATTTTCGT-GAATCCGGAGTGTAAGTAAAGTGTGGTGATGTTGA;14734_armA_-galK_R1_ATCGTCCGGGGCCAGCGCGGTGGTTTGCGTTAGCATTGT TGTCGTTTGTGAGCT. The DA14734/pSIM10-hygro culture was grown until early exponential phase (OD$_{600}$ ~ 0.3) and the λ Red system was induced by 15 min incubation in a 42 °C shaking water bath. The cells were made electrocompetent by washing 4-5 times in sterile 10% glycerol and 100-500 ng of the purified *armA(galK)* was electroporated. The cells were recovered in LB overnight at 30 °C. Successful transformants carrying *galK::armA* were selected on LA with gentamicin (50 mg/L) at 30 °C.

### Competitions during gastrointestinal colonization in mice

To assess the competitive advantage during gastrointestinal colonization in mice, the marked index isolate DA73939 (*galK::armA*) was competed with the unmarked selected isolates. Testing all isolates in mice was not feasible; therefore, we chose five isolates: DA14742, DA25061, DA25066, DA25071, and DA25116, representing different

isolation sites, capsule status, biofilm formation capacity, virulence in *G. mellonella* larvae, and survival in human serum. 6-8 weeks old BALB/c female mice ($n = 36$) were acclimatized for eight days before starting the experiment. One mouse injured itself during this period and had to be sacrificed by cervical dislocation. As a result, one competition group (control DA14734 vs DA73939) had five instead of six mice. Overnight cultures of pair-wise isolates were mixed 1:1 in PBS/1% bicarbonate, and 100 μl of the mixture (approximately $3 \times 10^7$ CFU) was administered via oral gavage. The procedure was repeated on two consecutive days (at least 24 h apart). On Day 5, a third gavage with fresh bacterial cultures was given. After that, drinking water with clindamycin (Villerton) at a final concentration of 0.5 g/L and 5 g/L glucose was provided *ad libitum* to disrupt the endogenous microflora and facilitate colonization. Faeces from each mouse were collected every day (Day 1-Day 10), weighed, homogenized in 1 ml sterile PBS, vortexed at full speed for 2 min, and briefly (max 5 s) spun down to remove larger debris. Different dilutions were plated on LB agar with cefotaxime (10 mg/L) to allow growth of all isolates and LB agar with gentamicin (50 mg/L) to select for the marked index isolate. The number of bacteria was expressed as CFU per gram of faeces. On Day 10, all mice were sacrificed by cervical dislocation. To test for systemic spread of bacteria, spleens were extracted, homogenized in 1 ml PBS, vortexed at full speed for 2 min, briefly spun (5 s), and plated on LB agar with cefotaxime (10 mg/L) or gentamicin (50 mg/L). Colonized mice showed no signs of disease or systemic infection at the end of the competitions. The competitive index was calculated each day by dividing the output ratio in faeces by the input ratio (inoculum cultures) for each mouse. The final data was normalized to the values from the control competition to account for any possible selective differences conferred by *galK::armA* and expressed as a log$_{10}$ competitive index.

### Growth under iron-depleted and repleted conditions

Iron-dependent growth in all isolates was assessed similarly as described before[49]. The iron-depleted medium was made by treating M9 with Chelex 100 (50–100 dry mesh) resin (Bio-Rad) for 1 h, then filter-sterilized and supplemented with 0.2% glucose, 1 mM MgSO$_4$ and 0.3 mM CaCl$_2$. Overnight cultures were grown in BHI and diluted to $10^3$ CFU/ml in M9/Chelex, M9/Chelex supplemented with 100 μM FeSO$_4$, and M9/Chelex with 5% heat-inactivated human serum. The growth was monitored in BioScreen C (Oy Growth Curves Ad Ltd) for 18 h at 37 °C with shaking and OD$_{600}$ measurements taken every 4 min. The index isolate was monitored along with the experimental isolates for each run. The area under the curve (AUC) was calculated for each isolate and each replicate and then normalized to the index for each condition. AUC of each isolate was compared to the index isolate in the respective conditions by Brown-Forsythe and Welch ANOVA followed by Dunnett's T3 multiple comparison test.

### RNA extraction and qPCR

Overnight cultures of selected isolates were grown in BHI and diluted to $10^4$ CFU/ml in M9/Chelex, M9/Chelex supplemented with 100 μM FeSO$_4$, and M9/Chelex with 5% heat-inactivated human serum. The cultures were grown until the mid-exponential phase, spun down, and mixed with two volumes of RNA Protect (Qiagen). RNA was extracted using the RNeasy Mini Kit (Qiagen), and genomic DNA was removed using the TURBO DNA-free kit (Invitrogen). Complementary DNA (cDNA) was synthesized using the High-Capacity Reverse Transcriptase kit (Applied Biosystems) with approximately 500 ng of extracted RNA. The mRNA levels of *fepA* and *fes* were quantified by qPCR (Illumina Eco Real-Time PCR System), using the SYBR Green Master mix (Thermo Fisher) with the following primers: fepA_F1 CAGAACACCAACTCCAACGA; fepA_R1 CCGTTCCAGGTAACCGAGTA; fes_F1 AATCGCTGCAACGTCTCC, fes_R2 TCACGGTCGGAGGGAATA. The mRNA levels of the housekeeping gene *glnA* were assessed using primers glnA_F and glnA_R[98]. The mRNA levels were normalized to the

index mRNA levels for the respective conditions using the $2^{-\Delta\Delta C_T}$ method[99].

## Sedimentation assay

Mucoviscosity was assessed by low-speed centrifugation as described before[40]. Overnight cultures grown in BHI (5 ml, in 50 ml Falcon tubes) were centrifuged for 5 min at 1000 x $g$, and $OD_{600}$ values were measured, before and after centrifugation, for the top 300 µl of supernatants. Mucoviscosity was expressed as the ratio between the $OD_{600}$ of the supernatant and the $OD_{600}$ of the corresponding culture before centrifugation.

## Statistical analysis

All statistical analysis was done using GraphPad Prism version 9.2.0. for macOS (GraphPad Software, San Diego, California, USA). The respective analysis is specified for each experiment and figure in the figure legends. Biological replicates in all experiments denote independent cultures grown from single colonies.

## Ethics statement

The study complies with all relevant ethical regulations. All experiments with animals were done in accordance with the national regulations and approved by the Regional Ethics Review Board in Uppsala (Dnr 5.8.18-15552/2019). All bacterial isolates were from routine samples and do not fall under ethical restrictions. Permission to include basic patient details linked to bacterial isolates was approved by the Regional Ethics Review Board in Uppsala (Dnr 2017/160).

## Reporting summary

Further information on research design is available in the Nature Portfolio Reporting Summary linked to this article.

# Data availability

The sequencing data generated in this study have been deposited at NCBI under bio project number PRJNA857654. Source data are provided with this paper and via figshare (https://doi.org/10.6084/m9.figshare.28678271). Biological material is available from the corresponding author upon reasonable request.

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

## Acknowledgements

We thank Evelina Kess for handling the work with mice during the colonization experiments, and the staff at the clinical microbiology lab at Uppsala University Hospital for help with the collection of clinical isolates. We also thank Dan I. Andersson and Vaughn S. Cooper for comments on the original manuscript draft, and Lionel Guy for help with the phylogenetic and gene enrichment analyses. All laboratory procedures were performed in BioSafetyLevel 2 (BSL2) facilities at Uppsala University, Sweden. No redactions of the results have been made.

## Author contributions

G.Z. and L.S. conceived the study and designed experiments. G.Z. performed all phenotypic characterization experiments. K.H. performed whole-genome sequencing. G.Z., K.H., and L.S. performed bioinformatic analysis. B.L. provided clinical isolates and patient data. G.Z. and L.S. analyzed data and wrote the manuscript. All authors read, edited, and approved the final version of the manuscript.

## Funding

## Competing interests

The authors declare no competing interests.
