## [Transparent Peer Review file · Nature Communications]

Parallel within-host evolution alters virulence factors in an opportunistic *Klebsiella pneumoniae* during a hospital outbreak

Corresponding Author: Professor Linus Sandegren

Version 0:

Reviewer comments:

Reviewer #1

(Remarks to the Author)

This is an interesting study of the evolution of an outbreak clone of *Klebsiella pneumoniae* within a hospital that documents a striking amount of parallelism in the locus-level genetic targets of mutations between isolates recovered from different patients. The dataset is unique and the general approach to the analysis is logical. This study nicely complements similar findings from patients with *K. pneumoniae* bloodstream infection and other gram negative infections. The investigation is thorough, yet the findings are not particularly novel or unexpected based on prior literature. I do have a few concerns about the statistical analyses, data presentation, and terminology used, which I think should be addressed. Also, I found the data to be overanalyzed in some instances and the manuscript to be presented in an overly complex format, which made it difficult to read. However, all of my comments are minor and could be easily addressed without further experiments.

1. Statistics. It would be more rigorous to analyze the gene mutation dataset (Fig 4d and 2e) using a gene set enrichment analysis (GSEA) and gene enrichment analysis (GEA). This enables one to ascertain whether or not the number of mutations observed in each category (gene function category or specific gene) is more than what would be expected by chance alone. I suspect that some of the strongest signals will be statistically significant and the main conclusions will stand. Nevertheless, this level of rigor is now becoming standard in the field. I know of one reference that supplies the R code for this analysis with instructions for how to do it:

Elgrail MM, Chen E, Shaffer MG, Srinivasa V, Griffith MP, Mustapha MM, Shields RK, Van Tyne D, Culyba MJ. Convergent Evolution of Antibiotic Tolerance in Patients with Persistent Methicillin-Resistant *Staphylococcus aureus* Bacteremia. *Infect Immun*. 2022 Apr 21;90(4):e0000122. doi: 10.1128/iai.00001-22. Epub 2022 Mar 14. PMID: 35285704; PMCID: PMC9022596.

2. Statistics. The binomial test is frequently deployed to test the hypothesis that 'infection' isolates are more enriched in some gene mutation categories than the 'colonization' isolates. First, I'm not sure that a statistically significant binomial test is biologically meaningful when deployed for these data in this manner. The mutations are parsed into multiple categories prior to testing a single category. Second, even if this is the correct statistical test, the P-values are not adjusted for multiple comparisons. Overall, I suspect this statistical treatment is overly sensitive to minor differences, none of which are likely to be biologically meaningful. What is more striking about these data is that fact that the same loci and genes in the same functional categories were mutated in both the 'infection' and 'colonization' isolates. In fact, from a biological viewpoint, the distributions of mutated gene categories between 'infection' and 'colonization' isolates would appear to be extremely similar, rather than different. I think that is the take-home point for me. They are very similar. The statistical differences noted between 'infection' and 'colonization' by binomial test are likely trivial. Consultation with a statistician may be warranted for a proper statistical assessment of this particular 'infection' vs 'colonization' enrichment question.

3. Terminology. This is a semantic issue, but an important one. I think using the term "infection" to classify all non-rectal swab isolates is misleading. From a clinical standpoint, classifying these isolates as 'infection' is likely incorrect in a substantial fraction of cases. Positive urine cultures do not necessarily equal an infection and may instead represent either 'asymptomatic bacteriuria' or colonization of indwelling urinary catheters. This is important since 80% of the isolates are from the urine of hospitalized patients. A significant fraction of these urine isolates are probably not true infections. For figure labels, consider the more precise descriptors of 'surveillance rectal swabs' and 'clinical specimens' instead. In the Discussion, you should explicitly mention that you cannot distinguish between infection and colonization for most of the

'clinical specimens' given the above. However, you can certainly note that the observed mutations may still be relevant for infection: *Klebsiella* in the urine is ultimately sourced from the GI tract and colonization of indwelling urinary catheters is a known risk factor for UTI. The main mutation signals in the dataset seem most likely to represent adaptations to the GI tract since GI colonization precedes urine colonization and the mutations involved seem to be similar between GI tract and urine.

4. Data presentation. Fig 2 is overly complicated to no real benefit. Consider using just 3 moth killing phenotypes (same, faster, slower). Panel 2f is particularly awkward and should be simplified into a single scatter plot.
5. Association vs. causation. It should be more clearly pointed out in the Discussion that no genetic manipulations were performed to assess a causal link between any genotype and phenotype. This is important because many isolates had multiple mutations. It was pointed out that some isolates only differed by few mutations, but short-read WGS can still miss some types of mutations. Only the index isolate was sequence by both long and short read technologies. The gold-standard to determine a genotype-phenotype causal link is still experimental genetic manipulation.
6. Most of the supplementary files are not organized in a way that is particularly useful.
7. What is the sequence type (ST) of the *K. pneumo* strain causing this outbreak? What is the prevalence and geographical distribution of this ST? Also, according to the reference #19 and #20, this was a CTX-M15 ESBL-producing strain and a multiresistance cassette was transferred to this strain from *E. coli* ST131. Although these details are not critical for this paper's main findings, I think briefly mentioning this will provide valuable context for the study for more clinically-minded readers. Consider adding a very brief summary of this information in the introduction where these other studies are cited.
8. Lines 54-56: "Transmission of this outbreak clone occurred via direct person-to-person contact and not from contamination of hospital surfaces (19); therefore, genetic changes under positive selection observed among isolates should illustrate adaptive evolution to the host rather than to the environment."

Please modify this sentence as it is not entirely accurate. Hospitalized patients rarely directly contact each other even when sharing the same room. Reference #19 actually even explicitly states both "direct" and "indirect" person-to-person contact and seems to implicate poor hand hygiene of healthcare workers as a contributing factor. It is much more likely that transmission between patients occurred via "fomites" and other contaminated hospital surfaces that served as transmission vectors between patients. I think what you are trying to say is that the outbreak was driven by patient-to-patient transmission rather than by exposure to some larger, shared environmental reservoir, like a contaminated hospital water supply. I think your point about adaptation to the host is likely to be true, because your data support that (lines 71-74). However, this is the Introduction section of the manuscript and, a priori, it's possible that the genetic changes facilitate stability on fomites and other hospital surfaces thereby making patient-to-patient transmission more likely. For example, there are the infamous outbreak cases of contaminated endoscopes that served as vectors for CRE transmission, which is an "indirect" patient-to-patient transmission mechanism. One can certainly imagine how biofilms could play a role in facilitating an outbreak with this type of mechanism.

Consider rephrasing to: "An epidemiologic investigation showed the outbreak occurred via direct and indirect patient-to-patient transmission rather than from a common environmental source (19); therefore, any genetic adaptations that occurred are likely to facilitate either transmission or survival within the host."

9. Line 326: "In total, 320 patients were identified with the clone...". How was this identification done? Per reference #19, it was PFGE at the time of that study. This is a small, but important detail since these isolates were curated from a much larger and diverse collection of *K. pneumo* strains circulating in the hospital.

10. It is concerning that only a fraction of the stocked isolates could be recovered due to a freezer accident. Were any of the isolates included in this study also recovered from the broken freezer? Any concerns about this causing mutations?

11. Fig 1b: I don't understand why the 'inset' starting in 2006 is needed here for this calculation. Why wouldn't you include all the available isolates in each category? Also, these data are noisy (as expected) and the 'infection' isolates are underrepresented from 2008-2009. Are these estimated mutation rates statistically different from each other? Do you think the mutation rate difference is actually biologically meaningful or just noise?

12. Line 64: "Strong positive within-patient selection targets key virulence factors at infection sites". "Strong" compared to what? I would remove the adjective 'strong'. Also, "Within-patient": Are you sure? The data are suggestive, but the outbreak was likely severely under-sampled. How do you know the mutations didn't cause stability on fomites, facilitating transmission chains that were never picked up due to under-sampling? Maybe they were initially selected for outside the host. You could comment that your follow-up experiments 'suggest' they were selected for in vivo, but I'm not sure you know that based on just a phylogenetic tree.

13. Line 102: "Strong" selection. Strong compared to what? I don't think you have any idea what the selection coefficient is here. Also, the adjective "vital" seems odd, since the isolate was recovered. If the virulence factor was "vital" than you would not even be able to detect its loss because the isolate would have died. Please avoid superfluous adjectives.

14. Line 199-200. "There was no direct relationship between acute virulence features (*G. mellonella*, serum, 199 mucoviscosity) and biofilm capacity in these isolates (Fig. 4b)." Biofilm capacity is not a metric in Fig 4b, so unclear to me how Fig 4b makes this point. Consider testing for an association between 'moth killing' and 'biofilm capacity' only and putting that negative data in a supplemental figure.

15. Lines 202-206. The data for the biofilm phenotype of isolates DA25061 and DA25127 (Fig 4c) don't appear to be in Fig 4a. Are the CV-stained biofilm masses statistically different between the isolates annotated in Fig 4c?

16. Fig 4a. Please plot the biofilm mass metric on a log scale instead of breaking the axis.

17. Lines 222-224. Also, mice are not people, so would temper that statement.

18. Lines 299-306. This paragraph is odd. dN/dS cannot be applied to intergenic mutations. Everyone agrees that intergenic mutations can be important. That's not under dispute. Would just remove this paragraph.

19. Line 307. The adjective 'unprecedented' is not appropriate. Similar studies have been done with clonal bacterial outbreaks and phenotype follow-up and even taken the extra step to prove causal genotype-phenotype links by introducing mutations into isogenic strain backgrounds. Here is one example:

Copin R, et al. Sequential evolution of virulence and resistance during clonal spread of community-acquired methicillin-resistant *Staphylococcus aureus*. PNAS. 2019 Jan 29;116(5):1745-1754. doi: 10.1073/pnas.1814265116. Epub 2019 Jan 11. Erratum in: Proc Natl Acad Sci U S A. 2019 Feb 25; PMID: 30635416; PMCID: PMC6358666.

20. This study should be cited and discussed as it seems rather relevant:

Cheng S, Fleres G, Chen L, Liu G, Hao B, Newbrough A, Driscoll E, Shields RK, Squires KM, Chu TY, Kreiswirth BN, Nguyen MH, Clancy CJ. Within-Host Genotypic and Phenotypic Diversity of Contemporaneous Carbapenem-Resistant *Klebsiella pneumoniae* from Blood Cultures of Patients with Bacteremia. mBio. 2022 Dec 20;13(6):e0290622. doi: 10.1128/mbio.02906-22. Epub 2022 Nov 29. PMID: 36445082; PMCID: PMC9765435.

21. I think the evidence that certain mutations represent niche adaptations to certain anatomical sites is not particularly strong. The stats are weak and the phenotypic analysis doesn't really address this question head on.

Reviewer #2

(Remarks to the Author)

Zaborskytė et al. investigated the host adaptation of a *Klebsiella pneumoniae* clone during a five-year nosocomial outbreak at Uppsala University Hospital by combining genomics and phenotypic analysis. They observed mutations in virulence-associated genes and claimed that this led to repeated loss of acute virulence and changes in capsule, lipopolysaccharide, iron uptake regulation, and biofilm formation. They also hypothesized that adaptations were likely for specific niches like the urinary tract, with some trade-offs during gastrointestinal colonization.

While I find this study intriguing and acknowledge its importance in understanding *K. pneumoniae* evolution in vivo, there are significant issues that preclude its publication in its current form. Although the study has utilized some genomics investigations and phenotypic assays, critical analyses are lacking.

Major points

1. My primary concern is that the authors have not substantiated with molecular approaches whether the genomic changes, specifically in virulence-associated genes, observed in the different isolates are directly responsible for the associated phenotypes. While I understand the difficulty in preparing knock-outs in clinical *K. pneumoniae*, the current results appear speculative and hypothesis-driven, lacking concrete molecular validation and additional approaches such as transcriptomics. As a result, the findings remain descriptive at this stage.
2. The first point is associated to my second concern regarding the importance of expanding analyses to include transcriptomics (and proteomics) beyond initial qPCR assessments for selected genes. Such an approach would offer valuable insights into expression profiles and regulatory mechanisms at the transcriptomic level, thereby enriching our comprehension of the observed phenotypic changes and potential adaptive responses. It is widely recognized that evolutionary adaptation involves not only genomic changes but also intricate regulatory processes that are captured at the transcriptomic level.
3. The absence of additional background information in the main text regarding the clonal outbreak such as the specific *K. pneumoniae* pathotype involved, more information on the index isolate, and the timeframe of the individual isolates, makes it challenging to comprehend the transmission and evolution dynamics. Also, while the authors conducted phylogenetics based on the core genome, there is a lack of insight into the accessory genomes. Understanding the differences among isolates in this regard is crucial, considering that plasmids, including hybrid plasmids encoding important virulence features such as siderophores and hypermucoviscosity, are known contributors to bacterial virulence and resilience, and likely play a role in niche adaptation within the host.
4. The authors assert that transmission of this clone occurred through direct person-to-person contact, while also suggesting that the emergence of particular clonal lineages is unlikely. This implies that individual bacterial isolates (i.e., the index isolate, which seems unlikely) were potentially acquired by individual patients, either within the intestinal tract or directly at infection sites such as the urinary tract, and subsequently underwent evolution within each patient independently. As depicted in the phylogenetic tree, the observed differences (SNPs, deletions etc.) among isolates are not consistently numerous, underscoring the importance of considering this aspect. Moreover, it would be beneficial to compare isolates not only to the index isolate but also among each other, further elucidating the evolutionary dynamics and transmission routes within the outbreak.

Minor points

- Title and throughout: I find the term "convergent evolution" confusing. In the context of *K. pneumoniae*, it typically refers to the merging of traits from both hypervirulent and classical (MDR) pathotypes. I recommend providing clarification.
- Introduction: Likewise, the introduction lacks details on hvKp/cKp and significant virulence characteristics.
- Material/Methods and Results:
 - o The prevalence of urinary tract infections as the predominant infection site suggests a bias that needs to be corrected. It is likely that distinct bacterial pathways are responsible for various infection niches.
 - o Most phenotypic results miss statistical assessment/p-values.

- o There's too much speculation/discussion in the results section.
- o What exactly do the authors mean by bacterial "attenuation"? This should be explained scientifically.
- o What do the authors mean by "all of iron metabolism were targets"? I don't think that this is the case. The most important Enterobacterales siderophore-associated virulence traits such as *iroB* or *iucA* are not mentioned. Also, why, in your opinion, were these genes not mutated (or were they not present/were they not part of the analysis; this should be addressed)?
- o Bacterial serum assays and complement evasion assays are not the same, although they are related. I suggest rephrasing.
- o Although I appreciate the *Galleria* infection model for its advantages over the mouse model, as highlighted by the authors, it's important to acknowledge that there are differing opinions on this model that warrant consideration, especially regarding its suitability to assess *K. pneumoniae* virulence. Also, the authors should perform at least experimental triplicates. E.g., by Russo and MacDonald: "The *Galleria mellonella* Infection Model Does Not Accurately Differentiate between Hypervirulent and Classical *Klebsiella pneumoniae*".
- o How many times were the biofilm experiments performed?
- o To what extent does the mouse model accurately reflect the clinical colonization scenario experienced by patients, including factors such as antibiotic treatment regimens? This should be addressed.
- Discussion: Limitations of the study should be included in the discussion.

Reviewer #3

(Remarks to the Author)

The authors describe detailed genetic comparisons of 110 *K. pneumoniae* isolates associated with a single prolonged transmission chain within a single hospital. They examine mutations relative to an 'index' isolate (being the first identified in the hospital), and observe parallel mutations in common pathways/functional groups which they interpret as convergent evolution during infection. They explore virulence and biofilm related phenotypes of selected isolates (some phenotyping assays are done on all isolates), and try to use this to interpret the mutations in terms of specific adaptive traits. Overall it is very high quality work, and very interesting to see the combination of detailed genetic analysis with comprehensive phenotyping.

Major comments:

1. In some places the authors over-interpret the causative link between mutations and phenotypes. Unless one generates isogenic mutants, or compares natural isolates with closed genomes that differ by only a single mutation, one cannot attribute causality to a specific mutation, even if it is biologically plausible to do so. It is sufficient to say that the observed mutations are 'associated' with particular observed phenotypes and 'likely explain' them, however it is over-stepping to claim direct causative effects of individual mutations without proving this. The authors should review the language throughout to ensure that they don't over-interpret.
2. It is important to contextualise this study for those readers familiar with *Klebsiella pneumoniae* biology and pathogenicity, by describing early on the genetic characteristics of the outbreak strain. This is important for overall comprehensibility of the article, and to help readers to understand the potential generalisability of results across the broader *K. pneumoniae* population. What is its sequence type, K and O antigens, does it carry any acquired siderophores or the *rmp* hypermucoidy locus, does it carry the virulence plasmid? The authors discuss changes in iron metabolism including the core siderophore system enterobactin, but do not mention yersiniabactin (a very common acquired siderophore system present in up to 50% of clinical isolates) or other acquired iron loci; similarly they discuss hypermucoidy without clarifying whether the *rmp* locus is present or not.
3. The authors conduct many statistical comparisons, often with very small subgroup sample sizes, without any acknowledgement of or attempt to control for multiple hypothesis testing. Even within single groups of tests, there is not correction. For example, at Lines 111-113: One of 5 tests (Fig 2b) shows p-value of 0.024; however, because five tests were done, the adjusted p-value would be much higher and (depending on correction approach used) likely >0.05. While it is challenging to identify a specific solution to this given the range of different tests conducted, the authors really should (i) provide effect sizes (with 95% confidence intervals) rather than just p-values; and (ii) acknowledge this issue in the Discussion, and quantify the number of comparisons they have tested in order to help readers contextualise. E.g. a p-value of 0.02 is not so impressive given we have tested 100s of comparisons.

Minor comments:

1. The abstract mentions "phenotypic tests" but doesn't give any indication of the nature and scale of these. There are likely word limits on the abstract but it would be very helpful to inform potential readers that the paper contains data on mucoviscosity, serum survival, iron utilisation and *Galleria* infection model for all isolates, and additional phenotyping of biofilm formation and murine gut colonisation on selected isolates. This is a key strength of the paper.
2. The Introduction claims: "Transmission of this outbreak clone occurred via direct person-to-person contact and not from contamination of hospital surfaces;" – This statement is somewhat problematic. The cited paper claims "transmission occurred by direct and indirect patient-to-patient contact", but it is not clear on what basis the authors can really rule out any involvement of contaminated hospital environment or equipment. Further, the cited paper does not seem to cover the full duration of the sampling period from which isolates were drawn for the present study. As this debatable point about direct person-to-person contact is also not essential to the present study, it should be modified or toned down.

3. Line 69 states there is an “average rate of 3.0 mutations per genome per year” this seems to be based on a linear regression of the data shown in the scatter plot? What is the R^2 for these correlations? Overall using this data to estimate mutation rate is not convincing, the intercept is around 10, how are we to interpret this? Would one not expect 0-time difference to correspond to 0-genetic difference? Similarly, the authors quote separate mutation rates for infection and colonisation, apparently based on separate linear regressions for each subset; again, what is the R^2 for these lines? Is the difference in slopes significant? How do you interpret the different intercepts?

4. The phylogenetic tree reconstruction is problematic. In the Methods, the authors state that they inferred a maximum likelihood tree and then manually updated the branch lengths to reflect the number of changes from the index isolate. This is highly unorthodox and the use of manual ‘correction’ of a tree in this way raises the potential for human error.

i) How were mutations counted?

ii) How was the tree rooted? Is the index isolate placed at the root? (It is not labelled on the figure)

iii) How do you know the ‘index’ isolate truly represents the most recent common ancestor of all other isolates sequenced?

The Methods indicates the index isolate is the first isolate detected, from a patient returning from overseas and so presumably the assumption is this was the point of importation into the hospital? This rationale needs to be made explicit

iv) What measures were taken to check or test the accuracy of the tree after manually ‘updating’ the branch length? What tools and approaches were used to make these updates, and to verify them? One would usually expect this to be done algorithmically, e.g. inferring a minimum spanning tree or ML tree, and then using ancestral state reconstruction to map individual mutations onto branches.

5. Line 74 describes dN/dS as the “ratio of nonsynonymous vs. synonymous changes (dN/dS)” and the Methods confirms what is presented is the ratio of the counts of nonsynonymous and synonymous changes. This is not the definition of dN/dS, rather dN/dS (also known as Ka/Ks) is the ratio of the number of nonsynonymous substitutions per non-synonymous site vs the number of synonymous substitutions per synonymous site. There are different denominators for the number of possible nonsynonymous substitutions than synonymous mutations (NS sites vs S sites), as the latter are largely restricted to third-codon positions (ie there are far fewer, so you will see fewer S mutations even if they occur at the same rate-per-site as N, ie in the absence of purifying selection). The dN/dS ratios quoted in the text currently are therefore likely very inflated (by ~3x). See e.g. https://en.wikipedia.org/wiki/Ka/Ks_ratio.

6. How were the survival curves divided into 5 groups? Each curve was compared to the reference (index) isolate, but that leaves two groups - those that are significantly different from the index isolate and those that differ. How were those that differ assigned to four groups? It's not obvious that, if one plotted all the curves on the one plot, there would be a clear distinction into 5 clusters.

7. Line 128-103 – The definition of ‘hypermucoid’ is not clear here, nor could I locate it in Methods.

8. Line 137 - “In most cases, reduced killing of *G. mellonella* was coupled with decreased complement evasion (Fig. 2f)” - this interpretation is not obvious from Fig 2f, it looks like perhaps half of cases in the attenuated group are below the line $y=0$

9. Lines 177 - How many / which patients were actually catheterised?

10. Line 201-202: Fig 4(c) doesn't seem to have enough data to demonstrate that biofilm production is end-point adaptation.

11. Line 225 – 226 and Fig 5: It seems odd that a colonising isolate (DA25116) performed the worst in the colonisation assay. How do the authors explain this? Furthermore, with only 5 samples, which display wide variability, the causality is very tenuous.

12. Line 275 - discussion of loss of O antigen, please see recent paper “Carbapenem-resistant *Klebsiella pneumoniae* capsular types, antibiotic resistance and virulence factors in China: a longitudinal, multi-centre study” *Nature Microbiology* volume 9, pages 814–829 (2024) which notes similar findings (<https://www.nature.com/articles/s41564-024-01612-1>)

13. Lines 299 - When discussing the relevance of non-coding mutations to bacterial virulence,

(i) The authors may like to also mention the example of *Salmonella* ST313 (<https://pubmed.ncbi.nlm.nih.gov/29487214/>)

(ii) The authors should note that there are reported examples in *K. pneumoniae* that synonymous intragenic changes can be functionally significant

(e.g. <https://pubmed.ncbi.nlm.nih.gov/36095213/>)

Version 1:

Reviewer comments:

Reviewer #1

(Remarks to the Author)

I feel the authors were very responsive to the reviewers concerns and the revised manuscript is significantly improved. The clarity of the updated analyses and main findings have greatly increased the impact of this manuscript in my opinion.

This is a comprehensive analysis of the evolution of an outbreak clone of *Klebsiella pneumoniae* within a hospital that

documents a striking amount of parallelism in the locus-level genetic targets of mutations between isolates recovered from different patients. The dataset is unique and the approach to the analysis is logical. It distills an enormous amount of data into biologically relevant pieces. This study nicely complements similar findings from patients with *K. pneumoniae* bloodstream infection and other gram negative infections and also adds to the literature with a very robust dataset that includes a range of relevant phenotypic investigations. The investigation is thorough in its genetic and phenotypic analysis. Although genetic validation of genotype-phenotype associations was not performed, the dataset as a whole is compelling and offers a unique window between clinical infection features (e.g. UTI vs colonization) and potentially specific niche adaptations. I think this report will be of great interest to those who study within-host bacterial evolution as well as the specific virulence mechanisms, such as capsule biosynthesis and iron regulation.

Reviewer #2

(Remarks to the Author)

The authors have carefully addressed the majority of my previous comments, resulting in notable improvements to the manuscript. The revisions have enhanced the overall quality of the study. While the issue of the missing genetic manipulation or further analyses remains, as highlighted in my initial review, I do recognize that the authors have toned down the discussion on this and other points, while also providing a more thorough explanation of the limitations. Based on these improvements, I would recommend accepting the paper for publication.

We would like to thank all three reviewers for their thorough and professional reviews of the manuscript. They have raised good questions and provided very nice and helpful suggestions to improve the manuscript. We have now extensively reworked the manuscript to address their concerns with a number of additional analyses, gone back to the involved physicians for extended information about the details of the outbreak, and hope that this revised version now will be fit for publication in Nature Communications.

REVIEWER COMMENTS

Reviewer #1 (Remarks to the Author):

This is an interesting study of the evolution of an outbreak clone of *Klebsiella pneumoniae* within a hospital that documents a striking amount of parallelism in the locus-level genetic targets of mutations between isolates recovered from different patients. The dataset is unique and the general approach to the analysis is logical. This study nicely complements similar findings from patients with *K. pneumoniae* bloodstream infection and other gram negative infections. The investigation is thorough, yet the findings are not particularly novel or unexpected based on prior literature. I do have a few concerns about the statistical analyses, data presentation, and terminology used, which I think should be addressed. Also, I found the data to be overanalyzed in some instances and the manuscript to be presented in an overly complex format, which made it difficult to read. However, all of my comments are minor and could be easily addressed without further experiments.

1. Statistics. It would be more rigorous to analyze the gene mutation dataset (Fig 4d and 2e) using a gene set enrichment analysis (GSEA) and gene enrichment analysis (GEA). This enables one to ascertain whether or not the number of mutations observed in each category (gene function category or specific gene) is more than what would be expected by chance alone. I suspect that some of the strongest signals will be statistically significant and the main conclusions will stand. Nevertheless, this level of rigor is now becoming standard in the field. I know of one reference that supplies the R code for this analysis with instructions for how to do it:

Elgrail MM, Chen E, Shaffer MG, Srinivasa V, Griffith MP, Mustapha MM, Shields RK, Van Tyne D, Culyba MJ. Convergent Evolution of Antibiotic Tolerance in Patients with Persistent Methicillin-Resistant *Staphylococcus aureus* Bacteremia. *Infect Immun.* 2022 Apr 21;90(4):e0000122. doi: 10.1128/iai.00001-22. Epub 2022 Mar 14. PMID: 35285704; PMCID: PMC9022596.

- We are very grateful for this suggestion by the reviewer and for pointing out the publication with a script to perform the analysis. We have done the gene enrichment analysis as suggested both on coding and non-coding regions (new Fig 1) and used the results as a basis for further analysis of genotype-phenotype connections throughout the manuscript. Indeed, the same top genes came out as significant and the main conclusions stand.

2. Statistics. The binomial test is frequently deployed to test the hypothesis that 'infection' isolates are more enriched in some gene mutation categories than the 'colonization'

isolates. First, I'm not sure that a statistically significant binomial test is biologically meaningful when deployed for these data in this manner. The mutations are parsed into multiple categories prior to testing a single category. Second, even if this is the correct statistical test, the P-values are not adjusted for multiple comparisons. Overall, I suspect this statistical treatment is overly sensitive to minor differences, none of which are likely to be biologically meaningful. What is more striking about these data is that fact that the same loci and genes in the same functional categories were mutated in both the 'infection' and 'colonization' isolates. In fact, from a biological viewpoint, the distributions of mutated gene categories between 'infection' and 'colonization' isolates would appear to be extremely similar, rather than different. I think that is the take-home point for me. They are very similar. The statistical differences noted between 'infection' and 'colonization' by binomial test are likely trivial. Consultation with a statistician may be warranted for a proper statistical assessment of this particular 'infection' vs 'colonization' enrichment question.

- We have reevaluated the biological reasoning behind the approach as suggested by the reviewer and agree that the division of genes into functional categories have drawbacks that result in effects that may not be biologically relevant. Instead, we have now focused more on the enriched genes from the GEA since that analysis is much more stringent. We still find a significant bias in the accumulation of GEA mutations among clinical isolates when we consider combinations of the enriched genes. However, as the reviewer points out, the overlaps of mutations between colonizing and infecting isolates indicates the biologically relevant fact that the mutants, in many cases, still cause the typical cKp type of infections. We have extended the discussion around what this means with respect to the *K. pneumoniae* transmission/infection cycle.

3. Terminology. This is a semantic issue, but an important one. I think using the term "infection" to classify all non-rectal swab isolates is misleading. From a clinical standpoint, classifying these isolates as 'infection' is likely incorrect in a substantial fraction of cases. Positive urine cultures do not necessarily equal an infection and may instead represent either 'asymptomatic bacteriuria' or colonization of indwelling urinary catheters. This is important since 80% of the isolates are from the urine of hospitalized patients. A significant fraction of these urine isolates are probably not true infections. For figure labels, consider the more precise descriptors of 'surveillance rectal swabs' and 'clinical specimens' instead. In the Discussion, you should explicitly mention that you cannot distinguish between infection and colonization for most of the 'clinical specimens' given the above. However, you can certainly note that the observed mutations may still be relevant for infection: *Klebsiella* in the urine is ultimately sourced from the GI tract and colonization of indwelling urinary catheters is a known risk factor for UTI. The main mutation signals in the dataset seem most likely to represent adaptations to the GI tract since GI colonization precedes urine colonization and the mutations involved seem to be similar between GI tract and urine.

- We agree with the reviewer that asymptomatic colonization of the bladder is frequent, especially among elderly patients. We have consulted again with clinicians that were in charge during the outbreak, and they confirm that isolates that we have labelled "infection" were sampled due to clinical indications and identified as the likely cause of the symptoms by the treating physician, so isolates from asymptomatic urine colonization were not

included in this category. Therefore, we feel that the division between these isolates and the rectal swabs is relevant. We have changed the labelling to “clinical” and “colonizing”, as also used in ref 26 from the outbreak, and extended the description to better illustrate how the different isolates were grouped. We agree fully with the reviewer that GI colonization precedes UTIs, and that mutations occurring in the GI tract therefore can be detected in both groups, but by identifying differences between isolates that came from gut colonization and those that presented symptoms in other parts of the body we may identify mutations that were selected in the different environments. We still find that combinations of enriched mutations were predominantly found in clinical isolates and have reported this in the results and discussed its potential implications.

4. Data presentation. Fig 2 is overly complicated to no real benefit. Consider using just 3 moth killing phenotypes (same, faster, slower). Panel 2f is particularly awkward and should be simplified into a single scatter plot.

- We have completely remade this figure and split it into Fig 2 and Fig 3 to make it less complicated. The new Fig 2 includes the new division of virulence groups from the *G. mellonella* infections: 1) attenuated (previously attenuated and slower killing), 2) no change, and 3) increased killing, as suggested. The “different dynamics” group is mentioned in the text instead. An extended analysis of genotype-phenotype connections is now also included and mapped on the phylogeny in both Fig. 2 and Fig. 3.

5. Association vs. causation. It should be more clearly pointed out in the Discussion that no genetic manipulations were performed to assess a causal link between any genotype and phenotype. This is important because many isolates had multiple mutations. It was pointed out that some isolates only differed by few mutations, but short-read WGS can still miss some types of mutations. Only the index isolate was sequence by both long and short read technologies. The gold-standard to determine a genotype-phenotype causal link is still experimental genetic manipulation.

- We agree with the reviewer and have modified the methods, results and discussion to point out how we have identified associations and that additional studies are required to verify exact causal links. However, the repeated identification of mutations in the same genes strongly suggest them to be under selection and this is strengthened by nearest neighbour analysis where we can in several cases support the phenotypic changes. For this study, the large number of mutations in different genes/functional clusters, the need to genetically combine several mutational combinations, and a genetic MDR background that presents clear problems with genetic manipulations makes reconstructing all possible differences highly challenging. We are following up some specific phenotypes with additional experimental evolution setups and a wider genetic background to broaden the scope in subsequent ongoing studies. One such study on biofilm evolution is available on bioRxiv (doi.org/10.1101/2024.03.16.585345) and complements the findings of this manuscript.

- All mutations were manually verified in the WGS sequences, and we re-sequenced many isolates to verify changes. Of course, it is possible that additional changes exist in genomic regions that for some reason were hard to sequence.

6. Most of the supplementary files are not organized in a way that is particularly useful.

- We have now reorganized the files, revised the information about figures and tables, and added a description of all files for the reader to navigate easier. We have also deposited all the raw data (Source data) for easier access for the reader.

7. What is the sequence type (ST) of the *K. pneumo* strain causing this outbreak? What is the prevalence and geographical distribution of this ST? Also, according to the reference #19 and #20, this was a CTX-M15 ESBL-producing strain and a multiresistance cassette was transferred to this strain from *E. coli* ST131. Although these details are not critical for this paper's main findings, I think briefly mentioning this will provide valuable context for the study for more clinically-minded readers. Consider adding a very brief summary of this information in the introduction where these other studies are cited.

- We thank the reviewer for pointing this out and have now included that information both as part of the introduction and at the start of the Results section since we also included some novel results and a description of how we choose samples for this study. The ST16 group of *K. pneumoniae* has been studied by others and phylogenies including the genome of the outbreak strain have been published previously. We have added these references to the guide interested readers.

8. Lines 54-56: "Transmission of this outbreak clone occurred via direct person-to-person contact and not from contamination of hospital surfaces (19); therefore, genetic changes under positive selection observed among isolates should illustrate adaptive evolution to the host rather than to the environment."

Please modify this sentence as it is not entirely accurate. Hospitalized patients rarely directly contact each other even when sharing the same room. Reference #19 actually even explicitly states both "direct" and "indirect" person-to-person contact and seems to implicate poor hand hygiene of healthcare workers as a contributing factor. It is much more likely that transmission between patients occurred via "fomites" and other contaminated hospital surfaces that served as transmission vectors between patients. I think what you are trying to say is that the outbreak was driven by patient-to-patient transmission rather than by exposure to some larger, shared environmental reservoir, like a contaminated hospital water supply. I think your point about adaption to the host is likely to be true, because your data support that (lines 71-74). However, this is the Introduction section of the manuscript and, a priori, it's possible that the genetic changes facilitate stability on fomites and other hospital surfaces thereby making patient-to-patient transmission more likely. For example, there are the infamous outbreak cases of contaminated endoscopes that served as vectors for CRE transmission, which is an "indirect" patient-to-patient transmission mechanism. One can certainly imagine how biofilms could play a role in facilitating an outbreak with this type of mechanism.

Consider rephrasing to: "An epidemiologic investigation showed the outbreak occurred via direct and indirect patient-to-patient transmission rather than from a common

environmental source (19); therefore, any genetic adaptations that occurred are likely to facilitate either transmission or survival within the host.”

- We have changed the statement as suggested and extended the description of the proposed transmission routes (based on ref 26) to better describe this point. As seen from the epidemiological investigation of the outbreak, the clone was only found at two occasions on surfaces in the hospital, and the absolute majority of spread was identified as person-to-person contact. This included patients helping each other with personal things like getting food and personal effects in multi-bed rooms, and that more than half of the patients needed help with their personal hygiene and/or with feeding.

9. Line 326: “In total, 320 patients were identified with the clone...”. How was this identification done? Per reference #19, it was PFGE at the time of that study. This is a small, but important detail since these isolates were curated from a much larger and diverse collection of *K. pneumo* strains circulating in the hospital.

- PFGE was the method used at the time of the outbreak to identify that isolates did belong to the clone and to direct efforts of contact tracing to prevent further spread. Sweden is a low endemic area for resistant strains and at the time of the outbreak even more so. From ref 26 and new discussions with personnel who performed screening during the outbreak, it is clear that there were very few other ESBL-producing bacteria (both for *K. pneumoniae* and *E. coli*) circulating in the hospital at the time of the outbreak. Before 2005, there were a maximum of 15 ESBL-producing isolate identified per year at UUH, and during the screening period there were only 14 ESBL-producing *K. pneumoniae* isolates identified that did not belong to the clone (ref 26). This is a very small proportion and would not affect any conclusions in the study. We have added more of this to the Materials & Methods section. WGS confirmed the clonality of all 110 isolates that we have studied here.

10. It is concerning that only a fraction of the stocked isolates could be recovered due to a freezer accident. Were any of the isolates included in this study also recovered from the broken freezer? Any concerns about this causing mutations?

- The freezer incident did not affect any of the isolates included here as they had been transferred to our laboratory prior to that. The “incident” was that during a clearing of collected samples from the hospital freezers all the remaining isolates of the clone were thrown away even though they were supposed to be kept. We have updated the statement to better describe why no additional isolates can be included in the study. It would have been great to have the complete collection of isolates, as this would likely have made it possible to trace individual mutations and spread between patients at a much greater level.

11. Fig 1b: I don’t understand why the ‘inset’ starting in 2006 is needed here for this calculation. Why wouldn’t you include all the available isolates in each category? Also, these data are noisy (as expected) and the ‘infection’ isolates are underrepresented from 2008-2009. Are these estimated mutation rates statistically different from each other? Do you think the mutation rate difference is actually biologically meaningful or just noise?

- The screening did not start until 2006 as a measure to curb the outbreak, so there were only clinical isolates up until then. We have replotted the graph to avoid confusion. We have also reevaluated the analysis and, as pointed out by another reviewer, it makes a big difference if the regression is forced through 0 (as would be more biologically relevant) than if the line is aligned best to the rest of the data due to the large variation. We have therefore dropped the analysis.

12. Line 64: “Strong positive within-patient selection targets key virulence factors at infection sites”. “Strong” compared to what? I would remove the adjective ‘strong’. Also, “Within-patient”: Are you sure? The data are suggestive, but the outbreak was likely severely under-sampled. How do you know the mutations didn’t cause stability on fomites, facilitating transmission chains that were never picked up due to under-sampling? Maybe they were initially selected for outside the host. You could comment that your follow-up experiments ‘suggest’ they were selected for in vivo, but I’m not sure you know that based on just a phylogenetic tree.

- We have removed the adjective as suggested. Since every patient from 2006 and onwards was screened for the clone at admission and when leaving the hospital, we think that most transmission chains were identified. There was extensive environmental sampling of hospital surfaces, sinks, toilets etc during the outbreak and only on two occasions were the clone found on a surface (once it was on a phone that the patient used right after using the bathroom and that was sampled directly after use). Of course, we cannot prove conclusively that no selection took place outside of the patients and we have modified the text throughout to avoid overstating this.

13. Line 102: “Strong” selection. Strong compared to what? I don’t think you have any idea what the selection coefficient is here. Also, the adjective “vital” seems odd, since the isolate was recovered. If the virulence factor was “vital” than you would not even be able to detect its loss because the isolate would have died. Please avoid superfluous adjectives.

- Strong was used in relation to the common assumption that a dN/dS value >1 indicates positive selection for a genetic change. We agree that we have no possibility to measure individual selection coefficients during an outbreak. “Vital” has also been removed. We have moderated the use of the adjectives throughout the text but still consider a dN/dS of 50 and up to 10 recurring independent mutations of for some genes as an indication of strong selection for the genetic changes.

14. Line 199-200. “There was no direct relationship between acute virulence features (G. mellonella, serum, 199 mucoviscosity) and biofilm capacity in these isolates (Fig. 4b).” Biofilm capacity is not a metric in Fig 4b, so unclear to me how Fig 4b makes this point. Consider testing for an association between ‘moth killing’ and ‘biofilm capacity’ only and putting that negative data in a supplemental figure.

- The previous version of Figure 4 only had the isolates with increased biofilm capacity plotted as most isolates do not show any biomass on pegs at all (silicone) or very minimal amounts (silicone with fibrinogen). To avoid confusion around this, we have remade the figure completely (Figure 5) and included all isolates. We have plotted biofilm capacity vs

mucoviscosity (now Fig. 5c), and biofilm capacity vs serum sensitivity or larval health scores are now included as Supplementary figure 6, as suggested.

15. Lines 202-206. The data for the biofilm phenotype of isolates DA25061 and DA25127 (Fig 4c) don't appear to be in Fig 4a. Are the CV-stained biofilm masses statistically different between the isolates annotated in Fig 4c?

- As pointed out for the previous comment, DA25061 and DA25127 isolates were not in Fig. 4c previously because only those with increased biofilm capacity were shown there. In the new Fig 5a all isolates are plotted. We have also added direct comparisons between the clustering isolates that increased biofilm vs those that lost it in Fig. 5b to show this difference more clearly.

16. Fig 4a. Please plot the biofilm mass metric on a log scale instead of breaking the axis.

- We have replotted with a continuous scale in the new Fig 5.

17. Lines 222-224. Also, mice are not people, so would temper that statement.

- We have altered the statement to point out the difference.

18. Lines 299-306. This paragraph is odd. dN/dS cannot be applied to intergenic mutations. Everyone agrees that intergenic mutations can be important. That's not under dispute. Would just remove this paragraph.

- The statement has been removed.

19. Line 307. The adjective 'unprecedented' is not appropriate. Similar studies have been done with clonal bacterial outbreaks and phenotype follow-up and even taken the extra step to prove causal genotype-phenotype links by introducing mutations into isogenic strain backgrounds. Here is one example:

Copin R, et al. Sequential evolution of virulence and resistance during clonal spread of community-acquired methicillin-resistant *Staphylococcus aureus*. PNAS. 2019 Jan 29;116(5):1745-1754. doi: 10.1073/pnas.1814265116. Epub 2019 Jan 11. Erratum in: Proc Natl Acad Sci U S A. 2019 Feb 25;: PMID: 30635416; PMCID: PMC6358666.

- We are aware of this study and have altered the previous statement.

20. This study should be cited and discussed as it seems rather relevant:

Cheng S, Fleres G, Chen L, Liu G, Hao B, Newbrough A, Driscoll E, Shields RK, Squires KM, Chu TY, Kreiswirth BN, Nguyen MH, Clancy CJ. Within-Host Genotypic and Phenotypic Diversity of Contemporaneous Carbapenem-Resistant *Klebsiella pneumoniae* from Blood Cultures of Patients with Bacteremia. mBio. 2022 Dec 20;13(6):e0290622. doi: 10.1128/mbio.02906-22. Epub 2022 Nov 29. PMID: 36445082; PMCID: PMC9765435.

- We thank the reviewer for pointing to this interesting study. We have included these results in the discussion in relation to other recent studies where *wzc* mutations were found.

21. I think the evidence that certain mutations represent niche adaptations to certain anatomical sites is not particularly strong. The stats are weak and the phenotypic analysis doesn't really address this question head on.

- We have reevaluated the statistics regarding the clinical vs colonizing isolates and in many cases toned down the previous conclusions when statistics have not been solid. We still find cases where some mutations point to being preferentially selected and stated this and discussed what we mean.

Reviewer #2 (Remarks to the Author):

Zaborskytė et al. investigated the host adaptation of a *Klebsiella pneumoniae* clone during a five-year nosocomial outbreak at Uppsala University Hospital by combining genomics and phenotypic analysis. They observed mutations in virulence-associated genes and claimed that this led to repeated loss of acute virulence and changes in capsule, lipopolysaccharide, iron uptake regulation, and biofilm formation. They also hypothesized that adaptations were likely for specific niches like the urinary tract, with some trade-offs during gastrointestinal colonization.

While I find this study intriguing and acknowledge its importance in understanding *K. pneumoniae* evolution in vivo, there are significant issues that preclude its publication in its current form. Although the study has utilized some genomics investigations and phenotypic assays, critical analyses are lacking.

Major points

1. My primary concern is that the authors have not substantiated with molecular approaches whether the genomic changes, specifically in virulence-associated genes, observed in the different isolates are directly responsible for the associated phenotypes. While I understand the difficulty in preparing knock-outs in clinical *K. pneumoniae*, the current results appear speculative and hypothesis-driven, lacking concrete molecular validation and additional approaches such as transcriptomics. As a result, the findings remain descriptive at this stage.

- We are aware that the golden standard is to perform reconstructions to verify exact causal links and have modified the discussion to point out that additional studies are required for this. However, the repeated identification of mutations in the same genes strongly suggest them to be under selection and this is strengthened by nearest neighbour analysis where we can in several cases support the phenotypic changes. For this study, the large number of mutations in different genes/functional clusters, the need to genetically combine several mutational combinations, and a genetic MDR background that presents clear problems with genetic manipulations makes this highly challenging. We are following up some specific

phenotypes with additional experimental evolution setups and a wider genetic background to broaden the scope in subsequent ongoing studies. One such study on biofilm evolution is available on bioRxiv (doi.org/10.1101/2024.03.16.585345) and complements the findings of this manuscript.

- We hope that the reviewer appreciate that most within-host evolution studies only have genomic data and there are very few studies that do so many phenotypic tests with all isolates, not just selected ones.

2. The first point is associated to my second concern regarding the importance of expanding analyses to include transcriptomics (and proteomics) beyond initial qPCR assessments for selected genes. Such an approach would offer valuable insights into expression profiles and regulatory mechanisms at the transcriptomic level, thereby enriching our comprehension of the observed phenotypic changes and potential adaptive responses. It is widely recognized that evolutionary adaptation involves not only genomic changes but also intricate regulatory processes that are captured at the transcriptomic level.

- We agree that transcriptomics and proteomics would give important information about how the different genetic changes affect the global gene regulation in the cell. However, since many of the genetic changes involve systems that are regulated by temporal switches via environmental signals and conducted through global regulatory signals like c-di-GMP, a comprehensive understanding of genotype-phenotype regulatory changes would require not only testing of standard growth *in vitro* but would need comparison with biofilm growth conditions, the different iron conditions, and genetic reconstructions and knock-outs of particular key regulators. Just running a basic transcriptomics profile of the 110 isolates (standard triplicate of samples = 330) to lay the baseline for what the different mutational combinations have resulted in would be quite a costly experiment. We therefore went for targeted measures of expression and consider an expansion of the present expression measures by transcriptomics and proteomics to be better suited for specific in-depth follow-up studies where the genetics behind particular phenotypes can be scrutinized in detail.

3. The absence of additional background information in the main text regarding the clonal outbreak such as the specific *K. pneumoniae* pathotype involved, more information on the index isolate, and the timeframe of the individual isolates, makes it challenging to comprehend the transmission and evolution dynamics. Also, while the authors conducted phylogenetics based on the core genome, there is a lack of insight into the accessory genomes. Understanding the differences among isolates in this regard is crucial, considering that plasmids, including hybrid plasmids encoding important virulence features such siderophores and hypermucoviscosity, are known contributors to bacterial virulence and resilience, and likely play a role in niche adaptation within the host.

- This is an important point and we have now included an extended description of the outbreak with details including the sampling of isolates, pathotype and other details of the *Klebsiella* clone, how it relates to other ST16 isolates etc. We have also included additional data regarding the accessory genome which consists of a 220 kbp multiresistance plasmid and a 5 kbp cryptic plasmid. All genetic variations in these plasmids are now included. The clone belongs to the classical Kp and does not encode any of the virulence genes of hvKp.

Some isolates had picked up additional plasmids during the outbreak but no additional resistance genes or virulence genes and this is now described.

4. The authors assert that transmission of this clone occurred through direct person-to-person contact, while also suggesting that the emergence of particular clonal lineages is unlikely. This implies that individual bacterial isolates (i.e., the index isolate, which seems unlikely) were potentially acquired by individual patients, either within the intestinal tract or directly at infection sites such as the urinary tract, and subsequently underwent evolution within each patient independently. As depicted in the phylogenetic tree, the observed differences (SNPs, deletions etc.) among isolates are not consistently numerous, underscoring the importance of considering this aspect. Moreover, it would be beneficial to compare isolates not only to the index isolate but also among each other, further elucidating the evolutionary dynamics and transmission routes within the outbreak.

- The transmission of the clone during the outbreak is based on previous studies by contact tracing and patient screening and we have now related back to those studies and expanded upon this to make it clear that transmission was dominantly by direct and indirect transmission between people and not due to any common environmental source (like contaminated sinks etc). The phylogenetic relationship between the isolates during the outbreak is consistent with this. We do see clusters of isolates that illustrate transmission routes and the genetic changes that have happened and have expanded the analyses of repeatedly genetic changes to illustrate examples of evolutionary trajectories as suggested. The differences in number of genetic changes are, with a couple of possible exceptions, not surprising given the short evolutionary time frame and the seemingly strong selection for particular changes identified in individual genes. Regarding comparison of all isolates against each other, that is what the phylogenetic tree represents. The algorithm places each isolate in relation to the genetic similarity of all other isolates. The rooting of the tree with the index isolate was done based on the temporal identification of the isolates and contact tracing that was performed at the hospital to identify the chain of transmission. The phylogenetics fully support the index isolates being the first case and that the spread occurred by subsequent transmission between patients.

Minor points

- Title and throughout: I find the term "convergent evolution" confusing. In the context of *K. pneumoniae*, it typically refers to the merging of traits from both hypervirulent and classical (MDR) pathotypes. I recommend providing clarification.

- In evolutionary biology, the term "convergent evolution" is traditionally used to describe independent parallel evolution resulting in the same phenotype. The opposite is "divergent evolution" that, for example, result in speciation by evolution of different traits. Nevertheless, we do acknowledge that using a term "convergent" in the context of *Klebsiella* can be ambiguous and thank the reviewer for pointing this out. Therefore, we have changed the title to avoid this.

- Introduction: Likewise, the introduction lacks details on hvKp/cKp and significant virulence characteristics.

- This is a good point, and we have expanded this information on different Kp pathotypes and their virulence features to provide a basis for the results presented here.

- Material/Methods and Results:

o The prevalence of urinary tract infections as the predominant infection site suggests a bias that needs to be corrected. It is likely that distinct bacterial pathways are responsible for various infection niches.

- We agree fully that distinct bacterial pathways are responsible for various infection niches and that selection for particular phenotypes therefore should be niche specific, but we do not see this as a weakness or a bias that affects the conclusions. Instead, UTIs are the predominant kind of infection for classical *K. pneumoniae* and the fact that we have many isolates from UTIs gives us a better chance to get a strong statistical basis for adaptations in this niche. We state throughout the text that possible adaptations in clinical isolates are likely linked to UTIs.

o Most phenotypic results miss statistical assessment/p-values.

- We have expanded the statistical analyses and their descriptions throughout the manuscript.

o There's too much speculation/discussion in the results section.

- We have tried to reduce the discussion in the results throughout the manuscript and only kept those points that we feel are required for interpretations that help the reader to understand the rationale for experiments and follow-up sections in the study.

o What exactly do the authors mean by bacterial "attenuation"? This should be explained scientifically.

- We have included an explanation of the use of virulence "attenuation" as virulence loss as described in other previous papers discussing virulence evolution (e.g. Diard and Hardt, 2017. Evolution of bacterial virulence. FEMS Microbiology Reviews).

o What do the authors mean by "all of iron metabolism were targets"? I don't think that this is the case. The most important Enterobacteriales siderophore-associated virulence traits such as *iroB* or *iucA* are not mentioned. Also, why, in your opinion, were these genes not mutated (or were they not present/were they not part of the analysis; this should be addressed)?

- We have modified the statement to avoid confusion. We only discuss iron-related genes that were mutated among the isolates. *iroB* and *iucA* are acquired genes linked to the most

prevalent virulence plasmid found in hypervirulent *K. pneumoniae* and are not found in this classical *K. pneumoniae* strain. We have clarified this in the text.

o Bacterial serum assays and complement evasion assays are not the same, although they are related. I suggest rephrasing.

- We agree with the reviewer and have rephrased this. Since we used the control with the complement inhibitor polyanetholesulfonic acid we could link the effect to complement. We have clarified this in the text.

o Although I appreciate the *Galleria* infection model for its advantages over the mouse model, as highlighted by the authors, it's important to acknowledge that there are differing opinions on this model that warrant consideration, especially regarding its suitability to assess *K. pneumoniae* virulence. Also, the authors should perform at least experimental triplicates.

E.g., by Russo and MacDonald: "The *Galleria mellonella* Infection Model Does Not Accurately Differentiate between Hypervirulent and Classical *Klebsiella pneumoniae*".

- Just as the referred paper shows, cKp does not perform well in mice models (0% killing) while hvKp easily kills the mice. The difference in killing effect in *G. mellonella* is much smaller since also the cKp can cause lethal infections, and this is why we choose *G. mellonella* to test variation in virulence for the varying mutants in cKp. In the current study we are comparing both killing and disease progression (vital parameters of the larvae such as melanization and mobility) and when used to compare genetically closely related bacterial isolates this model can provide good and reproducible differences, especially when research-grade larvae are used like in this study.

- We initially performed 2 biological replicates (10 larvae for each replicate) for all 110 isolates. The majority of isolates had identical or nearly identical results for both replicates, those that were not clear were repeated with 2 more biological replicates (10 larvae for each replicate). 2 isolates that consistently had high variation are still depicted as high variation to avoid placing them in the wrong group. This has been clarified in Materials & Methods.

o How many times were the biofilm experiments performed?

- They were performed with at least 4 biological replicates (individual inoculum cultures started from independent colonies) separately for silicone only and silicone with fibrinogen. Isolates with more variation or increased biofilm capacity were verified with additional replicates. Also, the isolates that are in the end considered to have increased biofilm formation show clear and really high amounts of biofilm biomass (in many cases above 2 and up to 9 when absorbance of dissolved stain is measured) which is very high and visible by the naked eye, so we are confident that they represent a true phenotype. The isolates where only one replicate showed increase are most likely spontaneous mutants that we know from another study can arise in this clone and were not considered. This has been clarified in the text and figure.

o To what extent does the mouse model accurately reflect the clinical colonization scenario

experienced by patients, including factors such as antibiotic treatment regimens? This should be addressed.

- cKp gut colonization is well known to be weak in the presence of the microbiota, and the golden standard for colonization experiments is to disrupt it by antibiotics (well reviewed in reference 64). Especially the anaerobic microbiota seems to prevent cKp colonization and clindamycin has been shown to create a stable situation for competitions. Several studies have shown that antibiotic treatment is a risk factor for colonization with multiresistant cKp and we have added this to the discussion. Most of the patients affected by the Uppsala outbreak clone was on antibiotic treatment at the time of isolation or just before, and this has been added to the manuscript.

- Discussion: Limitations of the study should be included in the discussion.

- We have modified the discussion in several places to illustrate weaknesses and what should be followed up in subsequent studies.

Reviewer #3 (Remarks to the Author):

The authors describe detailed genetic comparisons of 110 *K. pneumoniae* isolates associated with a single prolonged transmission chain within a single hospital. They examine mutations relative to an 'index' isolate (being the first identified in the hospital), and observe parallel mutations in common pathways/functional groups which they interpret as convergent evolution during infection. They explore virulence and biofilm related phenotypes of selected isolates (some phenotyping assays are done on all isolates), and try to use this to interpret the mutations in terms of specific adaptive traits. Overall it is very high quality work, and very interesting to see the combination of detailed genetic analysis with comprehensive phenotyping.

Major comments:

1. In some places the authors over-interpret the causative link between mutations and phenotypes. Unless one generates isogenic mutants, or compares natural isolates with specific mutation, even if it is biologically plausible to do so. It is sufficient to say that the observed mutations are 'associated' with particular observed phenotypes and 'likely explain' them, however it is over-stepping to claim direct causative effects of individual mutations without proving this. The authors should review the language throughout to ensure that they don't over-interpret.

- We have now moderated the statements regarding putative links between mutations and phenotypes as suggested throughout the text. We have reanalysed the genetics behind possible connections and included more descriptions of how this was interpreted. We have also added statements in the discussion to indicate further that genetic reconstructions are needed to fully decipher genotype-phenotype connections.

2. It is important to contextualise this study for those readers familiar with *Klebsiella pneumoniae* biology and pathogenicity, by describing early on the genetic characteristics of the outbreak strain. This is important for overall comprehensibility of the article, and to help readers to understand the potential generalisability of results across the broader *K. pneumoniae* population. What is its sequence type, K and O antigens, does it carry any acquired siderophores or the *rmp* hypermucooidy locus, does it carry the virulence plasmid? The authors discuss changes in iron metabolism including the core siderophore system enterobactin, but do not mention yersiniabactin (a very common acquired siderophore system present in up to 50% of clinical isolates) or other acquired iron loci; similarly they discuss hypermucooidy without clarifying whether the *rmp* locus is present or not.

- We thank the reviewer for pointing this out and have now included much more information on how the outbreak clone relates to general Kp biology and the division between core and acquired virulence factors to contextualise the study for readers. The clone belongs to the classical pathotype and did not contain any horizontally acquired virulence genes commonly present among hvKp. The discussion around different iron related genes is based on what genes were mutated, but of course the lack of context to other genes made it harder for the reader to put this in perspective. This has now hopefully been clarified.

3. The authors conduct many statistical comparisons, often with very small subgroup sample sizes, without any acknowledgement of or attempt to control for multiple hypothesis testing. Even within single groups of tests, there is not correction. For example, at Lines 111-113: One of 5 tests (Fig 2b) shows p-value of 0.024; however, because five tests were done, the adjusted p-value would be much higher and (depending on correction approach used) likely >0.05. While it is challenging to identify a specific solution to this given the range of different tests conducted, the authors really should (i) provide effect sizes (with 95% confidence intervals) rather than just p-values; and (ii) acknowledge this issue in the Discussion, and quantify the number of comparisons they have tested in order to help readers contextualise. E.g. a p-value of 0.02 is not so impressive given we have tested 100s of comparisons.

- As the reviewer points out, it is challenging to find a perfect solution to the problem of limiting both Type 1 and Type 2 statistical errors. We have now extensively reworked our statistical analysis in accordance with comments from all reviewers and after consultation with statistically well-versed colleagues. Where we use pair-wise tests between groups (such as in the new Fig 2) we do not perform any correction for multiple testing since each test is completely independent of the others. When testing all 110 isolates against the index we have used Dunnett's T3 multiple comparison test as a post-hoc for ANOVA. In the new Gene Enrichment Analysis in Figure 1, we have used a strict Bonferroni adjustment. All tests and approaches are described in the Materials & Methods and in figure legends. We have added 95% confidence intervals for all parameters in all figures as suggested.

Minor comments:

1. The abstract mentions "phenotypic tests" but doesn't give any indication of the nature and scale of these. There are likely word limits on the abstract but it would be very helpful to inform potential readers that the paper contains data on mucoviscosity, serum survival, iron

utilisation and *Galleria* infection model for all isolates, and additional phenotyping of biofilm formation and murine gut colonisation on selected isolates. This is a key strength of the paper.

- We agree and have now modified the abstract to accommodate this important information. In addition, we have changed the figure on biofilm (Now Fig 5) and included all isolates since biofilm capacity in the different conditions was measured for all 110 isolates and not just a selected subset.

2. The Introduction claims: "Transmission of this outbreak clone occurred via direct person-to-person contact and not from contamination of hospital surfaces;" – This statement is somewhat problematic. The cited paper claims "transmission occurred by direct and indirect patient-to-patient contact", but it is not clear on what basis the authors can really rule out any involvement of contaminated hospital environment or equipment. Further, the cited paper does not seem to cover the full duration of the sampling period from which isolates were drawn for the present study. As this debatable point about direct person-to-person contact is also not essential to the present study, it should be modified or toned down.

- We have extended the description of the proposed transmission routes (based on ref 26) to better describe this point. As described in ref 26 from the epidemiological investigation of the outbreak, the clone was only found at two occasions on surfaces in the hospital, and the absolute majority of spread was identified as person-to-person contact. Reference 26 does not cover the whole timeline of the outbreak but includes the majority of isolates identified (248 of 320). The large number of different wards where patients were identified with the clone limits the risk of there being a specific environmental source at the hospital, and the chain of transmission that was identified could explain all transmissions as being between patients.

3. Line 69 states there is an "average rate of 3.0 mutations per genome per year" this seems to be based on a linear regression of the data shown in the scatter plot? What is the R^2 for these correlations? Overall using this data to estimate mutation rate is not convincing, the intercept is around 10, how are we to interpret this? Would one not expect 0-time difference to correspond to 0-genetic difference? Similarly, the authors quote separate mutation rates for infection and colonisation, apparently based on separate linear regressions for each subset; again, what is the R^2 for these lines? Is the difference in slopes significant? How do you interpret the different intercepts?

- We have reevaluated this and come to the same conclusion as the reviewer and therefore the calculation has been removed. We still include the plot of the isolates as it gives a sense of the density of sampling over the outbreak.

4. The phylogenetic tree reconstruction is problematic. In the Methods, the authors state that they inferred a maximum likelihood tree and then manually updated the branch lengths to reflect the number of changes from the index isolate. This is highly unorthodox and the use of manual 'correction' of a tree in this way raises the potential for human error.

- We can see how adjusting the branch lengths can be viewed as unorthodox, but this is mainly because most phylogenetic analyses deal with datasets with too many genetic differences between the samples for this to be appropriate or even feasible. When large genetic distances are analyzed approximations must be made. Here, we have a manageable number of mutations, and a common genetic ancestor from which the individual mutations can be plotted on the tree during a very limited time period. Since phylogenetic models like ML uses approximations based on the possibility for multiple changes at the same site, branch lengths will not be integers but fractions and when we mapped out all individual changes on the branches, we felt that this was not an accurate representation of the data. This simply means that in the adjusted tree every branch with the exact same number of mutations is of the same length. No changes to the tree topology were made.

i) How were mutations counted?

- All genetic differences compared to the index were recorded based on reference assembly of Illumina reads on the reference genome followed by analysis of SNPs, InDels, and genetic rearrangements as described in the Materials & Methods. All genetic changes are included in Supplementary Table 1 for individual isolates, and in Supplementary table 2 listed according to the reference position.

ii) How was the tree rooted? Is the index isolate placed at the root? (It is not labelled on the figure)

- This is correct. The description in Materials and Methods have been updated with this.

iii) How do you know the 'index' isolate truly represents the most recent common ancestor of all other isolates sequenced? The Methods indicates the index isolate is the first isolate detected, from a patient returning from overseas and so presumably the assumption is this was the point of importation into the hospital? This rationale needs to be made explicit.

- This assumption was made from the timeline of the outbreak and the chain of transmission studies done at the hospital, and this has been included in the description now.

iv) What measures were taken to check or test the accuracy of the tree after manually 'updating' the branch length? What tools and approaches were used to make these updates, and to verify them? One would usually expect this to be done algorithmically, e.g. inferring a minimum spanning tree or ML tree, and then using ancestral state reconstruction to map individual mutations onto branches.

- With the very small number of mutations in the dataset we could manually map all mutations on the branches of the ML tree initially produced and adjusted the branch lengths to the integer number in the tree file. We then manually verified that no changes had occurred to the distribution of nodes and that the branch lengths matched each other if they had the same number of mutations.

5. Line 74 describes dN/dS as the "ratio of nonsynonymous vs. synonymous changes (dN/dS)" and the Methods confirms what is presented is the ratio of the counts of

nonsynonymous and synonymous changes. This is not the definition of dN/dS, rather dN/dS (also known as Ka/Ks) is the ratio of the number of nonsynonymous substitutions per nonsynonymous site vs the number of synonymous substitutions per synonymous site. There are different denominators for the number of possible nonsynonymous substitutions than synonymous mutations (NS sites vs S sites), as the latter are largely restricted to third-codon positions (ie there are far fewer, so you will see fewer S mutations even if they occur at the same rate-per-site as N, ie in the absence of purifying selection). The dN/dS ratios quoted in the text currently are therefore likely very inflated (by ~3x).

See e.g. https://en.wikipedia.org/wiki/Ka/Ks_ratio.

- The reviewer is correct and we are sorry for the poor calculation. We have now recalculated the dN/dS based on the method by Lipman presented by Nei and Gojobori (Mol. Biol. Evol. 3(5):418-426. 1986). We choose a simplified method based on the very short time frame of the outbreak and did not adjust for multiple substitutions in the same site for that reason. Most methods to calculate dN/dS are developed to account for the difficulties to detect selection in very diverged sequences between different species and not, as the case here, in a single clone over a very short evolutionary time span. The short time span means that purifying selection will not have had much time to act and therefore a higher than expected dN/dS can result without positive selection. However, the very large dN/dS for genes with three or more mutations, and the fact that they were mutated multiple times independently, suggests that a strong positive selection is acting on these genes under the studied circumstances.

6. How were the survival curves divided into 5 groups? Each curve was compared to the reference (index) isolate, but that leaves two groups - those that are significantly different from the index isolate and those that differ. How were those that differ assigned to four groups? It's not obvious that, if one plotted all the curves on the one plot, there would be a clear distinction into 5 clusters.

- The original division was based not only on killing but also on the health indexes of the larvae. However, based on this comment and comments by the other reviewers we have reconsidered this and now divided the isolates based on the survival curves into attenuated (less killing) and increased killing groups in addition to the group that statistically did not differ from the index isolate. This new division is used throughout the analyses and did not affect the outcome of other conclusions. Instead of using health index to further divide the isolates, we now discuss particular cases where this has relevance.

7. Line 128-103 – The definition of 'hypermucoid' is not clear here, nor could I locate it in Methods.

Since the term "hypermucoid" has been tightly associated with rmp-mediated capsule increase and the classical "string test", we have chosen to modify our description of isolates with morphotype changes and instead use the term "more mucoid" or "increased mucoviscosity" and couple this to the sedimentation results.

8. Line 137 - "In most cases, reduced killing of G. mellonella was coupled with decreased complement evasion (Fig. 2f)" - this interpretation is not obvious from Fig 2f, it looks like

perhaps half of cases in the attenuated group are below the line $y=0$

- Figure 2 has been completely remade into two figures (Fig 2 and 3). However, the reviewer is correct about the inaccurate phrasing in the previous version. There were 24 attenuated isolates with significantly reduced survival in serum, 24 without change in serum survival and 2 isolates with increased survival in serum. This has been corrected.

9. Lines 177 - How many / which patients were actually catheterised?

- Unfortunately, we do not have that information on the individual patient level. However, they were mainly geriatric patients with multiple co-morbidities which makes it likely that most of them had urinary catheters at least some time during the time period when they were at the hospital. From the outbreak study (Ref 26) it was reported that 37% had permanent urinary catheters and 40% had incontinence pads. We have added some of the general patient descriptions to the start of the text for readers to better contextualise the patient cohort.

10. Line 201-202: Fig 4(c) doesn't seem to have enough data to demonstrate that biofilm production is end-point adaptation.

- Figure 4c demonstrates two examples where isolates have lost the biofilm phenotype through specific mutations compared to their close relatives within clusters. The statement about gain of biofilm production was related to figure 4a where individual isolates all over the tree gained this independently through the indicated genetic changes instead of transmission. This has now been clarified in the text to avoid confusion.

11. Line 225 – 226 and Fig 5: It seems odd that a colonising isolate (DA25116) performed the worst in the colonisation assay. How do the authors explain this? Furthermore, with only 5 samples, which display wide variability, the causality is very tenuous.

- The colonizing isolate DA25116 was neutral in the colonization in mice, it was the UTI isolate DA25071 that had a negative trend in the colonization. This could indicate that it has adapted to another niche, but we do agree that the dataset is very small and would require further study to clearly verify the causality. We have now extended this in the discussion.

12. Line 275 - discussion of loss of O antigen, please see recent paper "Carbapenem-resistant *Klebsiella pneumoniae* capsular types, antibiotic resistance and virulence factors in China: a longitudinal, multi-centre study" *Nature Microbiology* volume 9, pages 814–829 (2024) which notes similar findings

(<https://www.nature.com/articles/s41564-024-01612-1>)

- Thank you for pointing this paper out. We have included it in the discussion on capsule loss mutants.

13. Lines 299 - When discussing the relevance of non-coding mutations to bacterial virulence,

(i) The authors may like to also mention the example of *Salmonella* ST313

(<https://pubmed.ncbi.nlm.nih.gov/29487214/>)

(ii) The authors should note that there are reported examples in *K. pneumoniae* that synonymous intragenic changes can be functionally significant (e.g. <https://pubmed.ncbi.nlm.nih.gov/36095213/>)

- We have limited the initial discussion of the relevance of intergenic mutations.